# Assessing the Influence of Adverse Weather on Traffic Flow Characteristics Using a Driving Simulator and VISSIM

**Chen Chen [1], Xiaohua Zhao [1,\*], Hao Liu [2], Guichao Ren [1], Yunlong Zhang [3] and Xiaoming Liu [1]**

1   Beijing Key Laboratory of Traffic Engineering, Beijing University of Technology, Beijing 100124, China; chen-chen@emails.bjut.edu.cn (C.C.); rengc@emails.bjut.edu.cn (G.R.); liuxm@bjut.edu.cn (X.L.)
2   Beijing Transportation Information, Beijing 100161, China; hao.liu@bjjtw.gov.cn
3   Zachry Department of Civil Engineering, Texas A&M University, College Station, TX 77843, USA; yzhang@civil.tamu.edu
\*   Correspondence: xiaohuazhao618@gmail.com; Tel.: +86-010-67396075

**Abstract:** The occurrence of adverse weather exacerbates traffic flow conditions, often leading to severe traffic congestions. Many studies have been conducted based on field-collected data to obtain the effects of weather on traffic flow characteristics. However, there is a limitation for filed data-based studies, in that weather conditions and traffic conditions are both noncontrollable and nonrepeatable, making it difficult to comprehensively assess the influence of weather conditions, especially the rare extreme weather conditions, on traffic flow characteristics. This paper proposes to assess these effects with the combination of driving simulator and traffic simulation. A driving simulator can collect driving behavior by conducting weather-related driving simulation experiments, while a microscopic traffic simulation program can evaluate the changes in traffic flow characteristics by inputting driving behavior parameters coming from the driving simulator. The proposed method can overcome the limitation of the field data-based approach. In this paper, the structure of the assessment platform is introduced at first. Then a verification experiment is conducted to measure the influences of adverse weather conditions on traffic flow characteristics. The verification experiment results show that the influences of adverse weather on traffic flow characteristics have consistent tendencies with outcomes from previous research and demonstrate that the method is practicable for the analysis of the influence of weather on traffic flow characteristics. This paper provides a practical way to analyze the influence of weather on traffic flow from driving behavior's point of view.

**Keywords:** adverse weather; driving simulator; traffic simulation; traffic flow characteristics

## 1. Introduction

In recent years, the number of vehicles in big cities has been increasing rapidly in China. In 2017, this number reached 5.9 million in Beijing [1]. The increased vehicle number and travel creates significant challenges to the roadway network of the city and traffic congestion becomes commonplace. Furthermore, there are occurrences of adverse weather conditions such as rainfall, snow, fog and haze that further exacerbate the traffic conditions. In July 2012, the biggest rainstorm in 61 years in Beijing resulted in severe traffic gridlock. On 12 December 2012, a snowfall caused low visibility and icy pavement, which interrupted traffic citywide for hours [2]. On 20 July 2016, the degree of congestion rose by 20% in Beijing due to heavy rain [3]. Thus, an in-depth understanding of how adverse weather influences traffic flow is essential to traffic management strategies for major weather events.

Many studies have focused on the impacts of weather on traffic flow characteristics such as average speed, traffic volume and road capacity [4–10]. Smith, Byrne, et al. [4] investigated the impacts

of different levels of rainfall intensity on freeway capacity and operation speed based on historical data. The results showed that freeway capacity reduction caused by light rain and heavy rain were 4–10% and 25–30% respectively. The average speed dropped 5.0–6.5% under the effect of rain regardless of intensity. Agarwal, Maze et al. [5] quantified the impacts of rain, snow and various pavement surface conditions on freeway traffic flow using a database that included four years of traffic occupancy and weather data and two years of pavement surface condition data. The results indicated that heavy rain and heavy snow would reduce the capacity by 10–17% and 19–27%, and reduce the speed by 4–7% and 11–15% respectively. Roh, Sharma, et al. [9] investigated the impact of snow on daily traffic volume. It was found that snowfall of 10 cm caused a 25% reduction in the daily volume. Previous studies of weather effects on traffic were mostly based on the field-collected data, which had a limitation that the weather and traffic conditions were both non-controllable and non-repeatable, making it difficult to comprehensively assess the impacts of different weather conditions. Particularly, the less frequent and more severe weather events and their impacts on traffic were hard to assess because of the lack of field data.

It is well known that the change in traffic flow condition is often due to the change in drivers' behaviors, and the driving behavior often changes under adverse weather conditions. In this paper, the impacts of weather focusing on the root of the problem are assessed, i.e., driving behavior change in adverse weather by innovatively combining a driving simulator and traffic simulation to overcome the limitations of the field data-based approach. A driving simulator is used to obtain driving behavior parameters needed in traffic simulation program by conducting weather-related driving simulation experiments. Traffic simulation is used to evaluate the change of traffic flow characteristics induced by adverse weather and the accompanying driving behavior changes via inputting driving behavior parameters coming from the driving simulator. Compared with field data-based research, this combination can take full account of various weather conditions including extreme weather events. On the other hand, the impacts of different weather conditions on traffic flow can be evaluated in simulation

There have been many driving simulators-based studies of weather effect on driving behavior. Broughton, Switzer, et al. [11] built three scenarios that have different levels of visibility to reveal factors that govern car-following. The experiments showed that in foggy weather, drivers were separated into two groups: staying within or lagging beyond the visible range of the front car. Yamaguchi and Sakakima [12] used a driving simulator to analyze driving behavior on the snow-covered roads. They found that it was impossible for drivers to keep the car in the center of the road. Konstantopoulos, Chapman, et al. [13] found that rainy weather had significant effects on drivers' eye movement based on a driving simulator. By utilizing the driving simulator-based method, Brooks, Crisler, et al. [14] claimed that vehicle speeds did not slow down significantly until a dramatic reduction in visible distance caused by fog. It was also suggested that the lane keeping ability decreased only when visibility was less than 30 m. Yan, Li, et al. [15] designed three scenarios with different risk levels in a driving simulator to investigate the effects of fog on drivers' speed control. The results indicated that drivers would reduce driving speed and acceleration to achieve a lower driving risk. As to the road surface friction in the driving simulator, Groot, Ricote, et al. [16] found that different levels of tire grip influenced lane keeping and driving speed.

Separately, some studies used traffic data under the influence of weather to calibrate traffic simulation models, aiming at optimizing traffic management measures and analyzing the efficiency of new management strategies [17–21]. Asamer, Zuylen, et al. [17] calibrated VISSIM to achieve a good match between simulated and observed saturation flow rate and start-up delay at a signalized intersection. Four driving behavior parameters sensitive to snow intensity including deceleration, acceleration, desired speed and clearance distance were chosen in their calibration. Then a sensitivity analysis found the best combination of parameter values that could make a simulation model best match with reality. In the research of Hou, Mahmassani et al. [18], a two-regime Greenshields' model was calibrated using weather adjustment factors. Then, the calibrated model was used in an estimation

and prediction system. The results showed that the calibrated model was capable of capturing the weather effects more realistically than without weather integration. Khavas, Hellinga, et al. [19] identified 21 input parameters used in a VISSIM calibration, including 10 parameters in Wiedemann 99 car-following model and maximum deceleration, accepted deceleration, etc. Then, a sensitivity analysis reduced the number of impact parameters to 9. At last, validation was conducted and the results showed that the calibrated model performed well. Generally, the calibrations of traffic simulation models in previous research are mostly based on actual data or by the way of sensitive analysis. Although this approach demonstrates a certain level of success, the non-controllability and non-repeatability of both nature weather and traffic conditions are still tough problems.

From the literature review above, two things can be summarized as following.

- The driving simulator is effective in simulating the weather effects on driving behavior.
- Microscopic traffic simulation programs like VISSIM can be calibrated via inputting weather-sensitive driving behavior parameters to evaluate the changes in traffic flow.

The advantages of driving simulator and microscopic traffic simulation program are obvious. The former can obtain driving behaviors under different weather conditions and the latter can output the measures of traffic flow characteristics influenced by driving behaviors.

To summary, an in-deep and comprehensive analysis of the effect of adverse weather on traffic flow is essential to help to the countermeasures development. However, field data-based studies are limited by the uncontrollable nature of adverse weather, making accurate analysis difficult. This paper aims to propose a method of using the driving simulator and traffic simulation to assess the influence of weather conditions on traffic flow, overcoming the limitations of the actual data-based approach. Both the research of the impacts of weather and the effects of other factors on traffic flow can be studied using this proposed approach to have a deep analysis from the angle of driving behaviors.

## 2. Method

### 2.1. Combination of the Driving Simulator and Traffic Simulation

The structure of the combination is shown in Figure 1. Driving behaviors act as the connector, integrating the two parts into one.

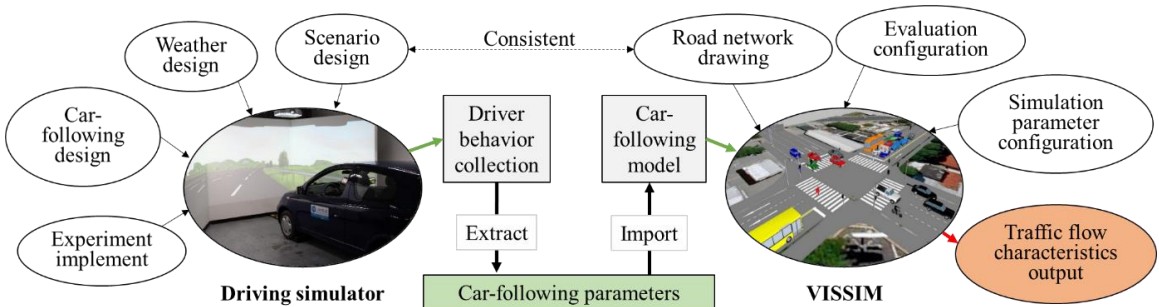

**Figure 1.** Combination of the driving simulator and simulation software.

The driving simulator used in this paper is constructed by INNO Simulation Company (Seoul, Korea) and is placed at Beijing University of Technology. In the driving simulator, different road scenarios and weather conditions can be simulated. During driving simulation, the scenarios with different weather visual effects are projected onto four big screens around a modified vehicle. These screens can provide a 130° field of view in front and a 30° field of view in back. Both vehicle operation data (speed, acceleration, lateral position, *XY* coordinate etc.) and driving behavior (gas pedal, brake pedal steering wheel etc.) can be recorded with the frequency of 1~50 Hz.

VISSIM is chosen as the traffic simulation program in this research. In recent years, VISSIM has become increasingly popular throughout the world and has been widely proved appropriate

in various of transportation research fields [22,23]. As the most common behavior, car-following has direct effects on traffic flow and is the focus of this paper. In VISSIM, car-following behavior is depicted by a psycho-physical model. There are two variants of this model with different parameters: Wiedemann 74 model and Wiedemann 99 model. Wiedemann 99 model is chosen in this paper due to that it is more advanced and flexible than Wiedemann 74 [24]. Wiedemann 99 model has ten essential parameters named CC0 to CC9. The detailed meaning of each parameter can be found in VISSIM User Manual [25].

In the driving simulator, road scenario, weather, and car-following situation are designed at first and driving behavior data is collected during driving simulation experiments. Referring to the demand of Wiedemann 99 car-following model, the needed parameters are extracted from the collected data. In VISSIM, simulation road net that is consistent with that in the driving simulator is drawn and traffic simulation parameters are configured to support the running of traffic simulation and the output of traffic flow characteristics. In application, parameters of car-following model are extracted from driving simulation and are imported into VISSIM. Then traffic simulation is run and the influenced traffic flow indicators are obtained. In this way, the influences of adverse weather on traffic flow characteristics are measured.

In the rest of this paper, a verification experiment is introduced in detail to measure the effect of adverse weather on traffic flow characteristics and to verify the proposed approach.

### 2.2. Driving Simulation Experiment and Parameter Extraction

In this part, a driving simulation experiment utilizing the introduced driving simulator is conducted to obtain car-following behaviors. The experiment design and its implementation are presented in detail as follows. After the experiment, car-following behavior parameters for Wiedemann 99 model are extracted and the results of the extraction are provided.

### 2.2.1. Apparatus

This research used a fixed-base driving simulator located at Beijing University of Technology, imported from Korea, produced by INNO-Simulation Company. The simulator includes a modified car (replacing the original vehicle accessories with computers or dynamic sensors), control computers and video and audio devices. Driving circumstances are projected onto four large screens (three ahead of and one behind the simulator car) and are displayed on two small screens on both sides of the car as side mirrors. This driving simulator is controlled by an embedded software called SCANeR Studio that is also developed by the producer. Using this software, the driving scenario can be fully controlled to perform nearly-true driving experience. Besides, the software record driving behavior (e.g., gas pedal, brake pedal, steering wheel angle) and vehicle operation data (e.g., speed, acceleration, distance to lead/rear car, $X/Y$ coordinates) during experiments at 1–50 Hz.

The validity of this simulator in studying driving behavior has been verified in previous research [26–28]. A total of 250 drivers have participated in driving-behavior-related research on this driving simulator, and this simulator has also been evaluated through questionnaires. The average score of the reality of the driving simulator reaches 8 (1-not real at all to 10-very real). Thus, research based on this simulator is considered valuable.

### 2.2.2. Scenario Design

The simulated scenario is built referring to Beijing E. 2nd Ring Road (urban expressway) from Zuoanmen Bridge to Xizhimen Bridge with the length of about 10 km as shown in Figure 2. The simulation road has three lanes in each direction with the width of 4 m per lane. Other road parameter like the position of exit and entrance, the structure of bridges, and markings and signs in the simulator are also designed according to the actual road. There are seven interchange bridges (labeled with ①~⑦ in Figure 2) in the simulated scenario and in each interchange bridge section, the simulated road passes underneath with the same design parameters. Therefore, each interchange

section contains a downslope and an upslope (longitudinal grade is 1.5). The 5th bridge contains both curve and longitudinal slope so as not to be focused in this experiment. Roads sections between two interchange bridges are classified to be a basic segment. Thus, three road types are included in this scenario: basic segment, downslope and upslope.

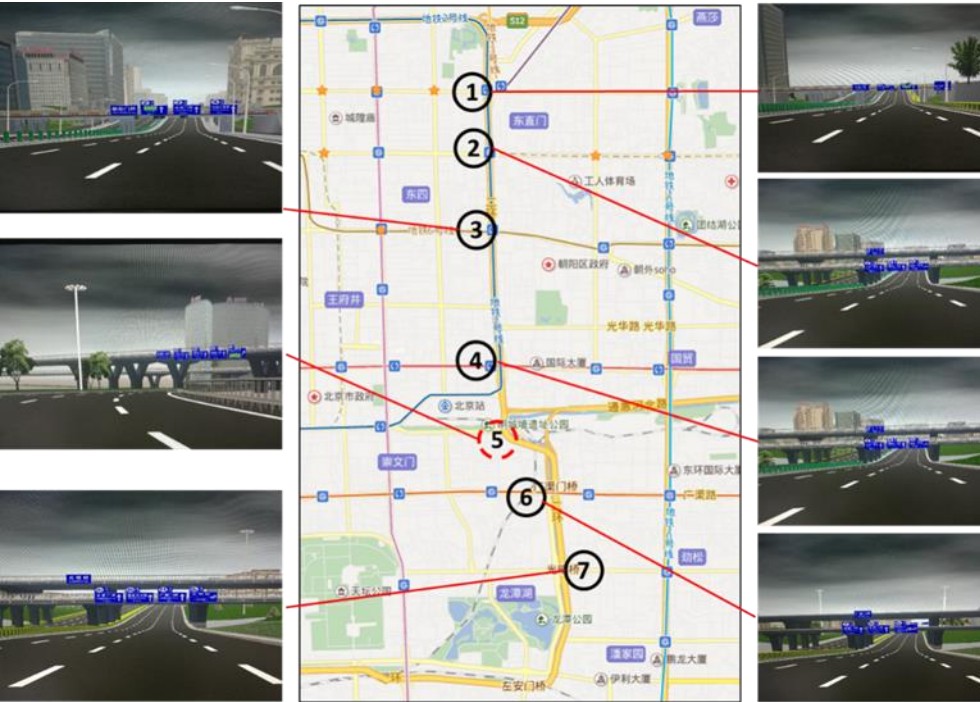

**Figure 2.** Overall view of simulation scenario.

### 2.2.3. Weather Design

In the driving simulator, there are three functions that can be used to simulate weather conditions: *SetRain*, *SetSnow* and *SetFog*. Among the three functions, *SetRain* and *SetSnow* are respectively used to control different intensities' visual effects of rain and snow. *SetFog* is used to control the visibility with the range from 0 m to 10,000 m (10,000 m is the default setting that means clear sky, i.e., the fog has no effect on drivers' visibility distance). The combinatorial configuration of the three functions allows the simulation of various weather conditions. The *SetRain* and *SetSnow* have the same value range of [0%, 100%], which represents the relative intensity of rain or snow instead of actual precipitation (*mm/12 h*). The values 0% and 100% stand for no rain/snow and the heaviest rain/snow respectively. Besides, the road friction (unit: %) that stands for the reduction coefficient relative to the sunny day, is automatically controlled by driving simulation software according to the values of *SetRain* and *SetSnow*.

In this research, 11 weather conditions are designed including a clear sky and ten adverse weather (4 levels of fog, 4 levels of rain and 2 levels of snow) and their configuration are shown in Table 1. To match the simulated weather conditions to actual weather grades, a 30-person participated visual experiment that aims at establishing the corresponding relationship between the driving simulation weather environment and actual weather grade that used in the weather grading in China. Students chose the most similar actual weather grade for every simulated weather condition after the observation of simulated weather. The matching results are shown in the column "Matched weather condition" in Table 1 as well. The visual effects of some of the 11 weather conditions are shown in Figure 3.

**Table 1.** Configuration of 11 weather conditions.

| ID | Configuration of Three Functions | | | | Matched Weather Condition (Actual Weather Grade) | Abbreviation |
|---|---|---|---|---|---|---|
| | *SetRain* (%) | *SetSnow* (%) | *SetFog* (m) | Friction (%) | | |
| 1 | - | - | 10000 | 100 | Clear Sky | CS |
| 2 | - | - | 1500 | 100 | Light Fog (1000 < S ≤ 100,000) | LF |
| 3 | - | - | 800 | 100 | Fog (500 < S ≤ 1000) | F |
| 4 | - | - | 300 | 100 | Dense Fog (200 < S ≤ 500) | DF |
| 5 | - | - | 50 | 100 | Heavy Dense Fog (50 < S ≤ 200) | HDF |
| 6 | 20 | - | 2000 | 100 | Light Rain (0–9.9) mm/24 h | LR |
| 7 | 45 | - | 800 | 75 | Rain (10.0–24.9) mm/24 h | R |
| 8 | 70 | - | 550 | 60 | Heavy Rain (25.0–49.9) mm/24 h | HR |
| 9 | 95 | - | 300 | 45 | Extremely Heavy Rain (100.0–249.0) mm/24 h | EHR |
| 10 | - | 45 | 500 | 45 | Snow (2.5–4.9) mm/24 h | S |
| 11 | - | 95 | 100 | 20 | Extremely Heavy Snow (10–19.9) mm/24 h | EHS |

*S: visibility distance.

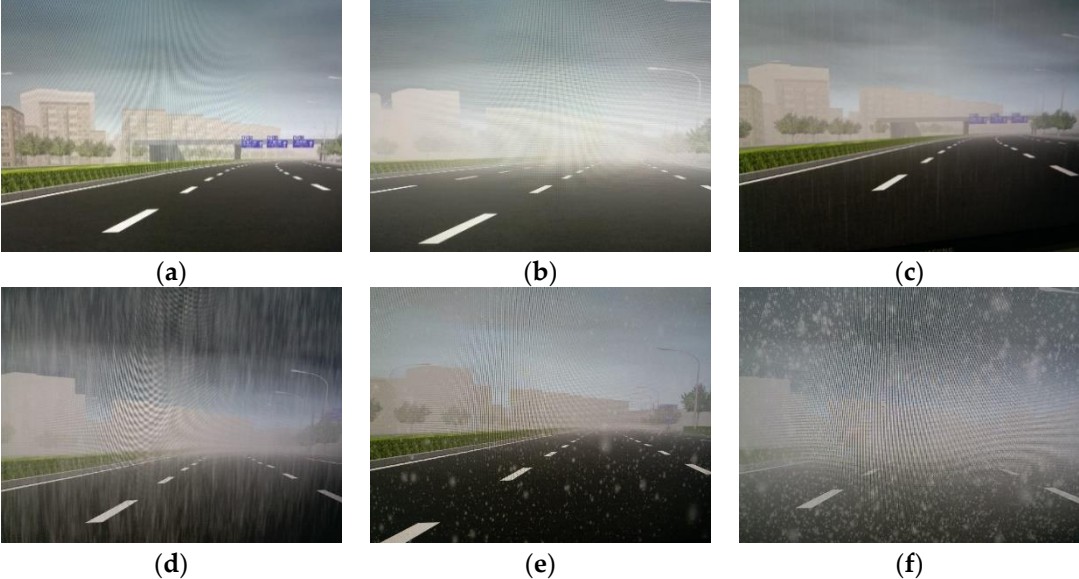

|  |  |  |
|---|---|---|
| (**a**) | (**b**) | (**c**) |
| (**d**) | (**e**) | (**f**) |

**Figure 3.** The visual effects of simulated weather. (**a**) Light Fog (**b**) Dense Fog (**c**) Rain (**d**) Extremely Heavy Rain (**e**) Snow (**f**) Extremely Heavy Snow.

### 2.2.4. Traffic Flow State and Car-Following Situation Design

The variation of traffic flow state in the driving simulator is configured by setting the speeds of surrounding vehicles, which is one of the functions of driving simulator software. Safety headway time of each surrounding vehicle is set as 2~3.5 s randomly. In this case, the combined configuration of speed and headway time can produce different traffic flow state. Three traffic flow states are designed in our experiments: (i) traffic flow with lower speed (average speed of surrounding vehicles is 40 km/h), (ii) traffic flow with higher speed (average speed of surrounding vehicles is 70 km/h) and (iii) free flow (speed limit of 80 km/h). The speed choice of 40 km/h is by reference to the average speed on Beijing

expressway in 2015 [29] and the speed choice of 70 km/h is based on the consideration of actual speed limit (80 km/h). Car-following situations are designed in traffic flow states (i) and (ii). The free flow scenario (iii) provides free flow traveling with few vehicles on road. The scenarios of traffic flow states (i) and (ii) are shown in Figure 4.

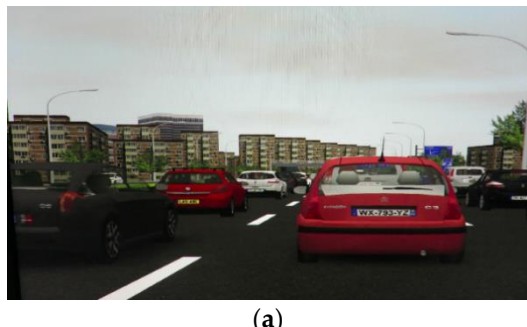 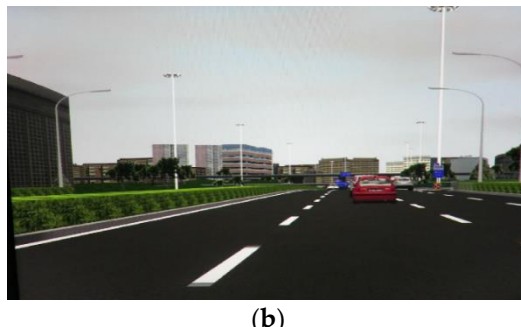

(**a**)          (**b**)

**Figure 4.** The visual effects of simulated weather. (**a**) Traffic flow (i); (**b**) traffic flow (ii).

In driving conditions, car-following situations are complex due to the variation of speed and acceleration of the front car, which make it impossible to simulate all of the car-following conditions in the driving simulator. Thus, three simplified categories of car-following situations are designed in our driving simulation experiment. The three car-following situations are cruising, accelerating and decelerating car-following situation respectively, according to front car's motion. After repetitive tests, car-following situations are well designed in the driving simulator and can give drivers similar driving experiences as the real world.

In the normal state, the front car is running at 40 or 70 km/h (for traffic flow state (i) or (ii) respectively).

- During cruising car-following process, the speed of the front car is the same as that in the normal state.
- During accelerating car-following process, there are three sub-processes. First, the front car accelerates to 50 or 80 km/h (for traffic flow state (i) or (ii) respectively) with a fixed acceleration of 3.333 or 5 km/h, lasting for 3 s. Then, the front car keeps the speed of 50 or 80 km/h for 2 s. At last, the front car returns to the normal state.
- During decelerating car-following process, there are three sub-processes. First, the front car accelerates to 30 or 55 km/h (for traffic flow state (i) or (ii) respectively) with a fixed acceleration of −3.333 or −5 km/h, lasting for 3 s. Then, the front car keeps the speed of 30 or 55 km/h for 2 s. At last, the front car returns to the normal state.
- Only the data during the first two sub-processes (5 s in total) is collected to extract parameters used in Wiedemann 99 car-following model. If car-following progress is interrupted (by lane-change or overtaking), the corresponding data will be discarded in the extract of the car-following parameter.

In traffic flow states (i) and (ii), the three car-following situations are randomly and uniformly allocated on three road types (basic segment, downslope and upslope). When the following vehicle (driver-controlled) passes through the certain locations, the front car will perform one of the predesigned motions described in previous.

2.2.5. Experiment Implementation

Thirty-one drivers (24 males and 7 females, based on the demographic characteristics of Chinese drivers [30], average age: $30 \pm 7.9$ years) with professional driving skills (average driving age: $15.8 \pm 6.9$ years) from a driving service company are recruited for this driving simulation experiment.

Once a driver arrives at the driving simulation laboratory, a pre-driving tutorial that lasts for approximately 10 min must be performed to help the driver adapt to the experiment equipment.

Then, the participant was asked to complete a questionnaire about the participant's basic information, including age, age of driving, time to fall asleep, time to wake up, and whether on medication or drunk. If a driver slept less than 6 h or took medicine or a drink during the past 12 h, his/her experiment would be canceled and adjusted to a later time.

For each driver, 11 weather conditions and 3 traffic flow states are randomly combined and sorted to eliminate familiarity, resulting in 33 different scenarios. Coupled with road type and car-following situation, an $11 \times 3 \times 3 \times 3$ (weather condition $\times$ traffic flow state $\times$ road type $\times$ car-following situation) designed experiment is performed. The average driving time for each scenario is about 7~10 min, followed by a rest lasts for about 3 min when the driving simulator loading next scenario. Each driver is arranged to complete their 33 times of driving in different three days, with each day driving 11 scenarios within about 2 h. During the driving experiment, participants are not allowed to use headlights when driving because that opened headlight will slightly increase the visibility distance, which may influence the desired setting of scenario. Driving behaviors data needed in the extraction of Wiedemann 99 model parameters like acceleration, distance to lead car and desired speed (obtained in free flow scenario) under various weather is collected with the sampling frequency of 20 Hz.

### 2.2.6. Parameters' Extraction

The 10 parameters used in Wiedemann 99 model are extracted according to their definition and the computational methods are listed in Table 2.

**Table 2.** Parameters used in Wiedemann 99 model.

| Parameter | Description | Computational Method |
|:---:|:---:|:---:|
| CC0 | Standstill distance | $CC0 = \sum_{n-1}^{N} d_{st,n}/N$ |
| CC1 | Headway time | $CC1 = \sum_{n-1}^{N} ((d_{cf,n} - CC0)/v_n)/N$ |
| CC2 | Following variation | $CC2 = \sum_{n-1}^{N} ((d_{cfma,n} - CC0 - CC1 \times v_n)/N$ |
| CC3 | Threshold for entering Following | $CC3 = \sum_{n-1}^{N} (t_{cfd,n} - t_{cf,n})/N$ |
| CC4 | Negative following threshold | $CC4 = \sum_{n-1}^{N} min(\Delta v_n)/N$ |
| CC5 | Positive following threshold | $CC5 = \sum_{n-1}^{N} max(\Delta v_n)/N$ |
| CC6 | Speed dependency of Oscillation | $CC6 = R_{\sigma v, d_{cf}}$ |
| CC7 | Oscillation acceleration | $CC7 = \sum_{n-1}^{N} |a_{cf,n}|/N$ |
| CC8 | Standstill acceleration | $CC8 = \sum_{n-1}^{N} a_{s,n}/N$ |
| CC9 | Acceleration with 80 km/h | $CC9 = \sum_{n-1}^{N} a_{cf80,n}/N$ |

Note: $N$: Number of experiment participants; $d_{st,n}$: Average distance to the front car when standstill of $n$th driver; $d_{cf,n}$: Average distance to the front car when car following of $n$th driver; $d_{cfma,n}$: Maximum distance to the front car when car following of $n$th driver; $t_{cfd,n}$: The time drivers start to decelerate before reaching safety distance of $n$th driver; $t_{cf,n}$: The time when reaching safety distance of $n$th driver; $\Delta v_n$: Speed difference between the following vehicle and front car when car following of $n$th driver; $R_{\sigma v, d_{cf}}$: Coefficient of correlation between $\sigma v_n$ and $d_{cf,n}$; $\sigma v$: Average of speed standard deviation of following vehicle when car following of all drivers; $d_{cf}$: Average distance to the front car when car following of all drivers; $a_{cf,n}$: Average acceleration when car following of $n$th driver; $a_{s,n}$: Acceleration when starting from standstill of $n$th driver; $a_{cf80,n}$: Average positive acceleration when speed is higher than 80 km/h of $n$th driver.

The ten parameters are classified into two groups due to their different data source: parameters that are independent of road types and car-following situations, and others that vary as the change of road types and car-following situations.

The first group includes CC0, CC3, CC8, and CC9. For CC0, it represents the headway at a standstill, and it is deemed to be the same no matter on slope or not. Thus, for the same weather condition, the same value of CC0 is used for all the three road types (basic segment, upslope and downslope). The reason is suitable for CC8 and CC9. For CC3, it is calculated during the process from the action of 'starting from parking' to the action of 'entering car-following' of the following vehicle, occurring at the beginning of the driving experiment. Another group includes the rest of the six parameters that will change under different weather conditions and road types.

Another consideration is how to process driving behavior data that comes from different traffic flow states and car-following situations in our driving simulation experiment. In terms of CC0, CC8, and CC9, it is easy to understand that they have nothing to do with traffic flow state, thus, they are not specifically designed in this experiment. For the same kind of weather condition, the average of data that is collected in all of the traffic flow states is used. Whereas, as to the rest of the seven parameters, the same set of values is used in VISSIM no matter what the car-following situation or traffic flow state are. Thus, for the same kind of weather, the average of data collected from all of the car-following situations and traffic flow states is used.

According to Table 3, the values of the ten parameters under different road types and weather conditions are listed in Table 3. The average desired speeds and their distribution under different weather situations are also obtained in the free flow scenario, due to the demand of VISSIM.

**Table 3.** Values of parameters extract from driving simulation experiments.

| Road Type | Weather Condition | CC0 (m) | CC1 (s) | CC2 (m) | CC3 (s) | CC4 (m/s$^2$) | CC5 (m/s$^2$) | CC6 (-) | CC7 (m/s$^2$) | CC8 (m/s$^2$) | CC9 (m/s$^2$) | Desired Speed (km/h) |
|---|---|---|---|---|---|---|---|---|---|---|---|---|
| Basic segment | CS | 4.45 | 0.87 | 5.28 | −7.92 | −1.52 | 1.52 | 0.71 | 0.31 | 1.03 | 0.33 | 67.93 |
| | LF | 1.84 | 1.51 | 6.38 | −8.57 | −1.26 | 1.26 | 0.73 | 0.32 | 1.30 | 0.30 | 72.26 |
| | F | 3.02 | 1.44 | 7.48 | −6.99 | −0.92 | 0.92 | 0.73 | 0.32 | 1.37 | 0.32 | 71.03 |
| | DF | 1.78 | 1.52 | 7.33 | −7.47 | −0.84 | 0.84 | 0.64 | 0.33 | 1.28 | 0.33 | 71.43 |
| | HDF | 1.60 | 1.26 | 19.40 | −5.37 | −0.83 | 0.83 | 0.25 | 0.34 | 1.18 | 0.26 | 53.63 |
| | LR | 9.54 | 1.21 | 6.36 | −7.47 | −0.67 | 0.67 | 0.64 | 0.34 | 1.30 | 0.35 | 72.48 |
| | R | 1.06 | 1.67 | 9.67 | −7.24 | −0.64 | 0.64 | 0.56 | 0.35 | 1.34 | 0.38 | 67.87 |
| | HR | 5.36 | 1.45 | 10.70 | −6.46 | −0.61 | 0.61 | 0.65 | 0.35 | 1.32 | 0.33 | 67.47 |
| | EHR | 1.34 | 2.31 | 11.20 | −8.39 | −0.60 | 0.60 | 0.68 | 0.36 | 1.34 | 0.31 | 71.11 |
| | S | 2.33 | 3.93 | 16.00 | −7.01 | −0.59 | 0.59 | 0.64 | 0.38 | 1.37 | 0.32 | 67.88 |
| | EHS | 1.00 | 10.88 | 20.00 | −8.09 | −0.43 | 0.43 | 0.57 | 0.39 | 1.36 | 0.30 | 63.9 |
| Upslope | CS | 4.45 | 1.30 | 8.58 | −7.92 | −2.10 | 2.10 | 0.62 | 0.40 | 1.03 | 0.33 | 67.93 |
| | LF | 1.84 | 1.24 | 8.89 | −8.57 | −2.36 | 2.36 | 0.60 | 0.37 | 1.30 | 0.30 | 72.26 |
| | F | 3.02 | 1.25 | 11.68 | −6.99 | −0.90 | 0.90 | 0.68 | 0.38 | 1.37 | 0.32 | 71.03 |
| | DF | 1.78 | 1.70 | 3.73 | −7.47 | −0.98 | 0.98 | 0.75 | 0.36 | 1.28 | 0.33 | 71.43 |
| | HDF | 1.60 | 1.18 | 20.70 | −5.37 | −1.15 | 1.15 | 0.50 | 0.36 | 1.18 | 0.26 | 53.63 |
| | LR | 9.54 | 1.08 | 13.33 | −7.47 | −1.41 | 1.41 | 0.68 | 0.38 | 1.30 | 0.35 | 72.48 |
| | R | 1.06 | 1.27 | 18.02 | −7.24 | −0.93 | 0.93 | 0.69 | 0.38 | 1.34 | 0.38 | 67.87 |
| | HR | 5.36 | 1.26 | 7.47 | −6.46 | −0.87 | 0.87 | 0.68 | 0.37 | 1.32 | 0.33 | 67.47 |
| | EHR | 1.34 | 2.36 | 19.43 | −8.39 | −1.02 | 1.02 | 0.57 | 0.36 | 1.34 | 0.31 | 71.11 |
| | S | 2.33 | 4.33 | 16.00 | −7.01 | −0.79 | 0.79 | 0.68 | 0.37 | 1.37 | 0.32 | 67.88 |
| | EHS | 1.00 | 6.74 | 20.00 | −8.09 | −1.06 | 1.06 | 0.47 | 0.40 | 1.36 | 0.30 | 63.9 |
| Downslope | CS | 4.45 | 0.56 | 3.84 | −7.92 | −2.53 | 2.53 | 0.63 | 0.42 | 1.03 | 0.33 | 67.93 |
| | LF | 1.84 | 0.82 | 4.64 | −8.57 | −2.19 | 2.19 | 0.54 | 0.47 | 1.30 | 0.30 | 72.26 |
| | F | 3.02 | 0.73 | 6.99 | −6.99 | −1.84 | 1.84 | 0.56 | 0.44 | 1.37 | 0.32 | 71.03 |
| | DF | 1.78 | 1.01 | 3.02 | −7.47 | −1.82 | 1.82 | 0.52 | 0.38 | 1.28 | 0.33 | 71.43 |
| | HDF | 1.60 | 0.67 | 11.91 | −5.37 | −1.83 | 1.83 | 0.42 | 0.41 | 1.18 | 0.26 | 53.63 |
| | LR | 9.54 | 0.57 | 6.31 | −7.47 | −1.73 | 1.73 | 0.55 | 0.41 | 1.30 | 0.35 | 72.48 |
| | R | 1.06 | 0.67 | 16.28 | −7.24 | −1.56 | 1.56 | 0.36 | 0.40 | 1.34 | 0.38 | 67.87 |
| | HR | 5.36 | 0.63 | 5.78 | −6.46 | −1.63 | 1.63 | 0.51 | 0.42 | 1.32 | 0.33 | 67.47 |
| | EHR | 1.34 | 2.00 | 14.13 | −8.39 | −1.76 | 1.76 | 0.47 | 0.45 | 1.34 | 0.31 | 71.11 |
| | S | 2.33 | 4.11 | 16.00 | −7.01 | −1.68 | 1.68 | 0.52 | 0.43 | 1.37 | 0.32 | 67.88 |
| | EHS | 1.00 | 8.14 | 20.00 | −8.09 | −1.48 | 1.48 | 0.38 | 0.46 | 1.36 | 0.30 | 63.9 |

*2.3. Traffic Simulation Based on VISSIM Software*

Using VISSIM software, a series of traffic simulations are conducted by inputting car-following driving behavior parameters listed in Table 3 to measure the effects of adverse weather on traffic flow characteristics.

2.3.1. Base Map Design

As the foundation of traffic simulation, the base map is drawn referring to Beijing E. 2rd Ring Road too. Due to the road net length limitation of VISSIM used in this research, only four interchange

bridges (labeled red ③~⑥ in Figure 5) are reproduced in VISSIM. The base map includes the main alignment of Beijing E. 2rd Ring Road and the other roads connecting to it. Divided by interchange bridges, the main alignment of the expressway is separated into five road sections (labeled black 1 ~ 5 in Figure 5).

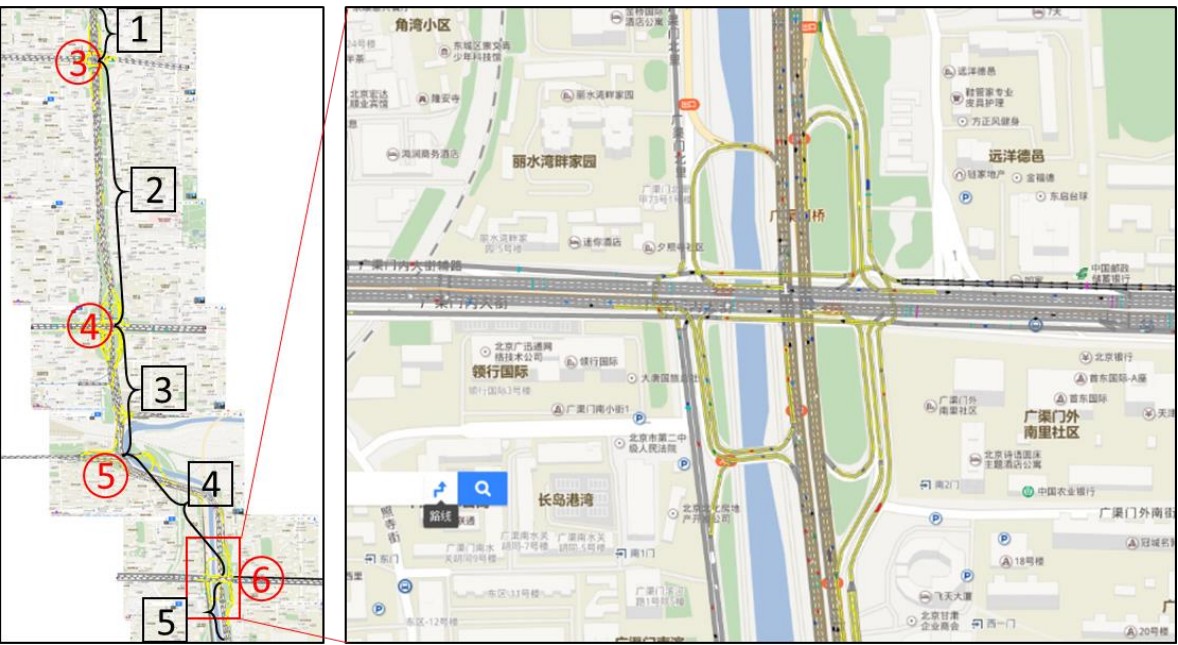

**Figure 5.** Base Map in VISSIM.

Two additional configurations are needed before traffic simulation: traffic volumes on the simulated road and traffic flow distributions in every merge or diverge section (entrance, exit, ramp etc.). Supported by traffic management department, data of traffic flow volume and speed in April 2016 is obtained. However, the traffic flow distributions are still unknown. Thus, some experience-based traffic flow distribution settings are assigned in the beginning and then the settings are adjusted in the calibration test.

2.3.2. Calibration of Traffic Flow Distribution

As described above, traffic flow distributions in the merge and diverge sections are needed to be calibrated to have the simulation to be matched with actual traffic state. The target parameter used in calibration is average speed on every road section. First, data of 9:00~10:30 am in clear sky are used to adjust the traffic flow distribution parameters and then data in light rain weather are used to verify the calibration. The results are shown in Figure 6.

As it is shown in Figure 6a,b, with the exception in 1 and 5 road sections, field speed and simulation speed on 2 3, 4 road sections have similar trends in both north to south and south to north directions. Although the differences between field speed and simulation speed for each road section are about 15–20 km/h, the consistent tend of field speed and simulation speed also proves that this simulation environment can be used to analyze the relative changes of speeds or other traffic flow characteristics under different weather conditions. The tendencies in Figure 6c,d that plot the field and simulation speeds of 2 3, 4 road sections in light rain condition, also prove the effectiveness of the trend. The exception that the 1st and 5th road sections have different trends between field speed and simulation speed is due to the reason that there are vehicle inputs on their beginning (i.e., the

initial point of vehicle generation). The effect can be seen at the [1] (the northernmost) road section in the north to south direction and the [5] (the southernmost) road section in the south to north direction. Therefore, in the following simulation, only the data of [2] [3], [4] road sections are collected.

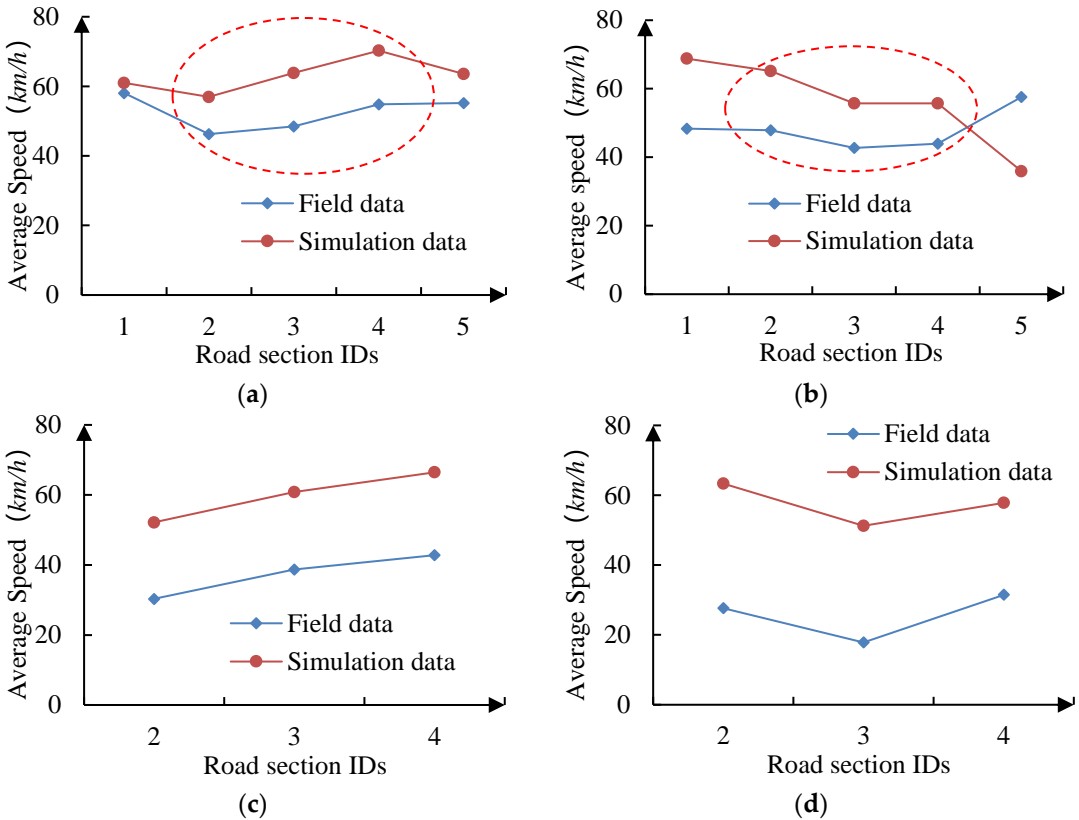

**Figure 6.** Calibration results. (**a**) Clear sky-North to South; (**b**) Clear sky-South to North; (**c**) Light rain-North to South; (**d**) Light rain-South to North.

### 2.3.3. VISSIM Simulation in Adverse Weather

First, the ten parameters of Wiedemann 99 model and the desired speed distributions are imported into the calibrated VISSIM simulation environment. VISSIM allows different driving behavior on different road types. Therefore, different road types (basic segment, upslope and downslope) are assigned with corresponding behavior parameter value settings. The traffic flow volume data used in simulation comes from the field data of 9:00~10:30 am in April 2016. Then a series of traffic simulations are conducted to obtain the traffic flow characteristics under various weather conditions. Besides, a sensitivity test is performed so as to better analyze the influence of weather on road capacity.

## 3. Results

The traffic flow characteristics and road capacity under various weather conditions are plotted in Figure 7. It should be noted that the labeled values in the figure represent the relative reduction percentages on the basis of the values of CS instead of the actual values of each indicator, to reflect the reductions percent of each traffic characteristic directly. as shown in Equation (1):

$$\text{reduction percentage} = \frac{value_{adverse\ weather}}{value_{Clear\ Sky}} \times 100\% \tag{1}$$

here: $value_{adverse\ weather}$ means the values of traffic flow characteristics of adverse weather. $value_{Clear\ Sky}$ means the corresponding traffic flow characteristics of clear sky.

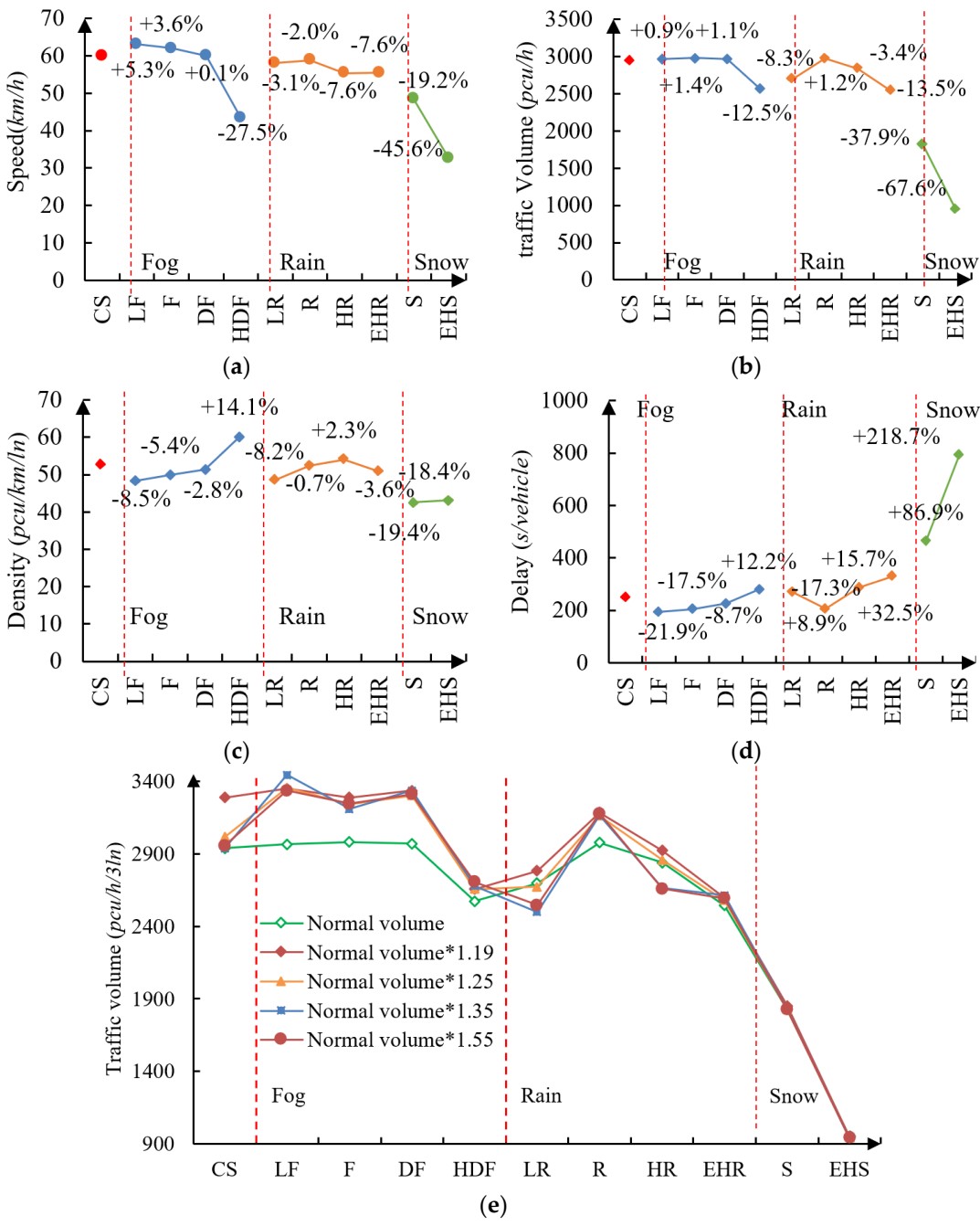

**Figure 7.** *Cont.*

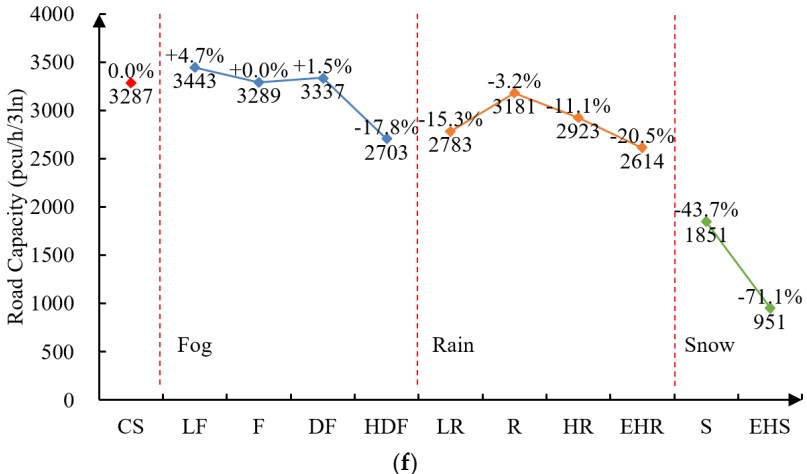

**Figure 7.** Traffic flow characteristics and road capacity under various weather. (**a**) Average speed, (**b**) traffic volume, (**c**) density, (**d**) delay, (**e**) sensitivity test results, (**f**) road capacity.

As can be seen in Figure 7a–d, average speeds in low-level fog and rainy weather have slight increases, which may be caused by the similar trend of desired speeds collected from the driving simulation. Average speeds have obvious decreases (7.6~27.5%) only in extreme weather (HDF, HR, EHR). Snowy weather severely reduces average speed (19.2~45.6%). Traffic volume has a similar tendency while delay has a contrary tendency with average speed. Density performs increasing tendencies in fog and rainy weather. Whereas the densities are low in snowy weather, which may be caused by the large headways selected by drivers to avoid rear-end collision in snowy weather, due to the extreme reduction of both visibility distance and road friction.

Five different traffic volumes are tested to find the road capacity (the maximum traffic volumes among five test) under different weather (Figure 7e). Both the traffic volumes (numbers below the dot) and relative reduction percentages (numbers above the dot) that take the volume of CS as reference are labeled in Figure 7f. Similar to Figure 7b, road capacities of low-level rains are similar with CS; while in extreme weather, road capacity has an obvious reduction (11.1~20.5%). The reduction coefficients are extremely high in snowy weather (43.7~71.1%).

## 4. Discussion

In this research, only car-following behavior is considered. Lane-changing and overtaking behaviors are not tested. Whereas, car-following is the most common behavior and its changes have a huge effect on traffic flow. It will be better if the other two behaviors can be taken into consideration.

One of the issues of driving simulator-based studies is the validation of the driving simulator. To solve this problem, we have conducted some validation experiments [26–28] and require each of participant to complete a questionnaire used to evaluate the reality of our driving simulation. It is showed that results from our driving simulator are in accordance with that from filed test; and participants feel the driving experiences in our driving simulator is similar to the real world. Although driving behavior parameters are not directly used in the validation experiment, it is thought that the relative effectiveness (comparison between treated groups and control group) of our driving simulation is valid in analyzing driving behaviors.

Table 4 shows the comparison between the results in our paper and results in previous paper. It can be seen that results based on the proposed method are similar to that in previous research except for that the speed reduction in light rain and rainy weather is small. However, we argue that adverse weather have impacts on the traffic flow volume and the results in our paper are based on the volume in clear sky while results in previous works are based on a lower volume (field data-based research), resulting in the differences. Whereas, Table 4 can prove that the combination of driving simulator and traffic simulation is usefully in the effect assessment of weather on traffic flow.

**Table 4.** Comparison between results in our paper and results in previous paper.

| Weather | Literature | Changes on Capacity | | Changes on Speed | |
|---|---|---|---|---|---|
| | | Result in Our Paper | Results in Literature | Result in Our Paper | Results in Literature |
| Light rain | Rakha, Farzaneh et al. (2008) | −15.3% | −10∼−11% | −3.1% | −8∼−10% |
| Rain | Agarwal, Maze, et al. (2005) | −3.2% | −7∼−8% | −2.0% | −8∼−12% |
| Heavy rain | Smith, Byrne, et al. (2004) Agarwal, Maze, et al. (2005) | −11.1% | −4∼−10% −10∼−17% | −7.6% | −5.0∼−6.5% −4∼−7% |
| Snow | Roh, Sharma, et al. (2014) | −43.7% | −25% | −19.2% | - |
| Heavy snow | Smith, Byrne, et al. (2004) | | −25∼−30% | | - |
| | Agarwal, Maze, et al. (2005) | | −19∼−27% | | −11∼−15% |

In this paper, participants in the driving simulation experiment are professional drivers, which may lead to bias on the results. In future works or further application, various types of drivers need to be taken into consideration to have better results. These participants are chosen from a driver database established by previous driving simulation studies. Thus, these drivers don't have simulator sickness and can drive correctly; the driving behavior is suggested to be available in the analysis of car-following driving behaviors.

Some values in Table 3 seem to be odd, such as the high CC0 value for clear sky. It should be noted firstly that the values in Table 3 are extracted from the driving behavior data from the driving simulation experiment and no modification is applied. The lower values of CC0 for adverse weather may be caused by the complex effect of decrease of visibility distance and surface friction. Changes in other values may be also due to similar reasons. However, although these parameter values may be puzzling, the rationality of results in Figure 7 can prove that the values in Table 3 are also reasonable. Thus, the proposed method in this paper can also be seen as valuable. In future works, a careful analysis of each parameter should be conducted to have better results.

Another issue is the desired speed in Table 3. The desired speed of the clear sky is lower than some of the desired speed of adverse weather. We have analyzed the influence of weather conditions on desired speed. It is found that except for the values of HDF, there is no significant difference between the values of other weather conditions. The slight difference between the desired speeds might be caused by the complex effect of the visibility distance and road surface friction. This phenomenon is different with our intuition. However, the desired speed is tested in an ideal environment (road length is 10 km—only a few cars are on the road). The results may be reasonable. The desired speed is input into VISSIM and functions as the max speed when the driving environment is available. Thus, it may not have a huge effect on the results. In future works, a more carefully scenario design for free flow should be focused to eliminate this question.

In the calibration part, the target is to let the average speed of simulation are the same as the field data. This goal is not achieved in this research. It might be because when calibrating VISSIM, only traffic volume is used and no other traffic characteristics are available in our research. Although we have worked hard to calibrate the VISSIM map, there still some differences in Figure 6. However, a consistent tendency is obtained. In terms of the objective of this paper that is measuring the effects of various adverse weather on traffic flow characteristics, the relative changes could be convincing. Meanwhile, the trends in this research are in accordance with previous outcomes, which can also prove the relative effectiveness of the method that it can be used to measure the influence of weather on traffic flow. In future works, if the free flow scenario can be designed more carefully or if richer field data can be obtained to support a better base map drawing and a better calibration, the results would be more accurate.

The basic idea of this paper is to propose a basic method by combining driving simulator and traffic simulation software to assess the effects of weather on traffic flow. The utilization of the advantages of both driving simulator (micro-driving behavior generation) and traffic simulation

(evaluation on traffic flow level based on micro-driving behavior models) can break the limitation of field-data-based study so as to provide a platform for the integrated analysis of traffic flow from the perspective of driving behavior.

Except for inputting the model parameters, the API embedded in the driving simulator and traffic simulation software (for VISSIM, it's a COM interface) can also be used to support the method. This idea can be regarded as a general method used in studies on the changes in traffic flow states that are deduced by changes in driving behavior.

## 5. Conclusions

This paper proposes a combination of driving simulator and traffic simulation used to measure the effect of weather on traffic flow characteristics. By first establishing a framework of the combination then conducting a verification experiment that includes a driving simulation experiment and a series of traffic simulations, the effects of weather on traffic flow characteristics are measured. Based on the results of the verification experiment, the proposed method is proved practicable in analyzing the influence of weather on traffic flow characteristics.

A bridge between driving behavior and traffic flow is established. Similarly, this method can also be used in other research fields, such as the influence of driving while fatigued or under the influence of alcohol and their effect on traffic flow. Moreover, it also provides a new perspective for the understanding of the mechanisms of traffic flow changes.

**Author Contributions:** Conceptualization, X.Z., Y.Z. and X.L.; Methodology, C.C. and X.Z.; Validation, C.C.; Investigation, C.C. and G.R.; Resources, H.L.; Data Curation, C.C., H.L. and G.R.; Writing—Original Draft Preparation, C.C.; Writing—Review & Editing, X.Z., Y.Z. and X.L.

**Funding:** This study was supported by the National Natural Science Foundation of China project (61672067), and Science and Technology Program of Beijing (Z151100002115040).

**Conflicts of Interest:** The authors declare no conflicts of interest.

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
