# Peer review of "Assessing the Influence of Adverse Weather on Traffic Flow Characteristics Using a Driving Simulator and VISSIM"

_sustainability, doi:10.3390/su11030830_

Round 1

Reviewer 1 Report

This paper evaluated the impacts of adverse weather on traffic flow using driving simulator and micro-simulation. However, there are two many issues should be addressed. Here is the comments.

1. The introduction showed background of vehicles and adverse weather in Beijing. Did this paper focus on Beijing? If so, how do you design the driving simulation scenarios in Beijing? In other words, how do you differentiate the driving simulation scenarios in Beijing and that in other place? If not, a proper background should be given.

2. The introduction is not clear and might confuse readers. It is suggested that an introduction and a literature review should be given separately.

3. Since the simulation focused on urban expressway, Wiedemann 99  car-following model should be avoided. Wiedemann 74 is suggested in the study because it is suitable for urban traffic.

4. What is the difference of “intensities’ visual effects” and “visibility”?

5. The adhesion coefficient of wheels is an important factor in adverse weather study. However, this study highly neglected the adhesion coefficient. This should be enhanced in the revision.

6. What exactly the data is extracted from the driving simulation experiments? I don’t think the parameters in Wiedemann 99 model could be extracted directly from the experiments. If you used the equations in Table 2 to calculate the corresponding  parameters, what original data do you use? What is the samples of these data? What is the statistic description? Do you use the average value of other percentage value?

7. There is a big difference between the field speed and the simulation speed, although the treads are similar. It indicated that the calibration of traffic flow distribution was failed. A recalibration should be conducted to improve the results.

8. The paper needs a proofreading. There are lots of grammar error and typos.

Examples, driver behavior should be “drivers’ behavior” or “drivering behavior”

Page 4, line 71-74.  “… can on one hand ….., and on the other hand, …”

Page 2, line 27-28.  “This paper provides a practical way to analyze the influence on traffic flow of changing driver behaviors.”

Author Response

Thank you for reviewing my manuscript and the comments. We have revised our manuscript according to your comments.

This paper evaluated the impacts of adverse weather on traffic flow using driving simulator and micro-simulation. However, there are too many issues should be addressed. Here is the comments.

Question 1. The introduction showed background of vehicles and adverse weather in Beijing. Did this paper focus on Beijing? If so, how do you design the driving simulation scenarios in Beijing? In other words, how do you differentiate the driving simulation scenarios in Beijing and that in other place? If not, a proper background should be given.

Response: Thank you for this comment. The aim of this paper is to perform a method that uses the combination of driving simulation and traffic simulation to assess the influence of weather condition on traffic flow. The background in Introduction is based in Beijing. However, it is thought that the effects of adverse weathers on traffic flow will also happen in other cities. Besides, the selection of Beijing is because the scenario is built based on Beijing E. 2nd Ring Road, which is regarded as a case study to verify the proposed method. We have modified some sentences in Introduction (See line 151-158) and Conclusion (See 547-550) to emphasize the focus of this study.

“To summary, an in-deep and comprehensive analysis of the effect of adverse weathers on traffic flow is essential to help to the countermeasures development. However, field data-based studies are limited by the uncontrollable of adverse weathers, making the accurate analysis difficult. This paper aims to propose a method of using the driving simulator and traffic simulation to assess the influence of weather conditions on traffic flow, overcoming the limitations of the actual data-based approach. Both the research on the impacts of weather and the effects of other factors on traffic flow can be studied using this proposed approach to have a deep analysis from the angle of driver behaviors.”

“A bridge between driving behavior and traffic flow is established. Similarly, this method can also be used in other research filed such as the influence of fatigue driving or drunk driving on traffic flow. Moreover, it also provides a new perspective for the understanding of the mechanisms why traffic flow changes.”

Question 2. The introduction is not clear and might confuse readers. It is suggested that an introduction and a literature review should be given separately.

Response: Thank you for this comment. The structure of the introduction is:

1) Importance of analysis of the effects of weather on traffic flow (the first paragraph).

2) The field data-based studies and their shortcomings (the second paragraph).

3) The proposal of the combination method (the third paragraph).

4) Relative studies based on driving simulation (the fourth paragraph).

5) Relative studies based on traffic simulation (the fifth paragraph).

6) The advantage of the two tools (line 142-150).

7) Summary and the aim of this paper (the last paragraph).

The reason why we embed literature review into the introduction is based on the consideration of better expression on the shortcomings in previous studies and the usage of both driving simulator and traffic simulation. To better convey our purposed and objective, in the last of the Introduction, we have added the summary of introduction (See lien 142-158)

Question 3. Since the simulation focused on urban expressway, Wiedemann 99  car-following model should be avoided. Wiedemann 74 is suggested in the study because it is suitable for urban traffic.

Response: Thank you for this comment. According to the VISSIM User Manual, It’s true that Wiedemann 74 model is mainly suitable for urban traffic while Wiedemann 99 model is suitable for interurban (motorway) traffic. Parameters in Wiedemann 74 model include average standstill distance (ax), additive part of safety distance (bx_add), multiplicative part of safety distance (bx_mult). Then the distance d=ax+bx, bx = (bx_add+bx_mult*z)*sqr(v), where z is a value that distributed around 0.5 with std=0.15. Obviously, compared with the parameters used in Wiedemann 99, the bx_add and bx_mult don’t have physical meaning, making it difficult for the extraction based on driving behavior. Besides, the the bx_add and bx_mult is relative to the speed, which needs a huge experiment under different speed. On the other hand, parameters in Wiedemann 99 are more related to the actual driving data, easier to extract. The literature [24] states that Wiedemann 99 model is more advanced and flexible than Wiedemann 74, and this literature also uses Wiedemann 99 model to simulate urban roads. Thus, in this research, Wiedemann 99 model is selected.

Question 4. What is the difference of “intensities’ visual effects” and “visibility”?

Response: Thank you for this comment. In our driving simulator, the ‘intensity’ is used to describe the visual effect levels of rains, snow, while ‘visibility’ is used to describe the distance that drivers can see. For example, in Figure 3f, the Extreme heavy snow, the value of SetSnow determines the intensity of the snow, i.e. the numbers of snowflake in the screen. However, the visibility distance doesn’t change and drivers can see the far road or vehicles from the gap between snowflakes. The SetFog can low the visibility distance and restrain drivers’ sight. The combinatorial configuration of the SetRain, SetSnow, and SetFog finally results in a desired visual effect of different weathers.

Question 5. The adhesion coefficient of wheels is an important factor in adverse weather study. However, this study highly neglected the adhesion coefficient. This should be enhanced in the revision.

Response: Thank you for this comment. It is thought that the adhesion coefficient of wheels is represented by the road surface friction (See line 246-248). The road surface friction setting is limited by the driving simulator software and it will change as the change of the value of SetFog and SetRain. Due to the company who produces the driving simulator is professional in the development of driving simulator, it is thought that the setting of friction is rational.

Question 6. What exactly the data is extracted from the driving simulation experiments? I don’t think the parameters in Wiedemann 99 model could be extracted directly from the experiments. If you used the equations in Table 2 to calculate the corresponding  parameters, what original data do you use? What is the samples of these data? What is the statistic description? Do you use the average value of other percentage value?

Response: Thank you for this comment. It’s true that parameters in Wiedemann 99 model cannot be directly extracted from the driving simulation experiment. Data from the driving simulator is a time-series structure including speed, accelerating, coordinate, distance to other vehicles etc. By applying the equation in Table 2, parameters in Wiedemann 99 can be calculated. Only data in the 5 second (See line 305) of the car-following situation is used to extract these parameters. Due to sampling frequency is 20 Hz and there are 31 participants, the total data is 31(participants)*5s*20Hz*3(car-following situation)*3(traffic flow)*3(weather condition) for each weather condition. As been shown in Table 3, the average values are used for these parameters. We have added the Standard deviation in the bottom of this response letter.

Question 7. There is a big difference between the field speed and the simulation speed, although the treads are similar. It indicated that the calibration of traffic flow distribution was failed. A recalibration should be conducted to improve the results.

Response: Thank you for this comment. There do have some difference between the field data and simulation data. It might because when calibrating VISSIM, only traffic volume is used and no other traffic characteristics are available in our research. But the focus of this paper is to emphasize the combination of driving simulator and traffic simulation. Compared with results in previous research, the results in Figure 7 seem to be rational (See line 481-489). Thus, it is suggested that the proposed method is valid. The differences between field data and simulation data in Figure 6 can be eliminated by a more reasonable driving simulation experiment design and full calibration of traffic simulation if field data can be abundant. We have added some explanation on this issue, See line 516-527.

“Table 4 shows the comparison between the results in our paper and results in previous paper. It can be seen that the results based on the proposed method are similar to that in previous research except for that the speed reduction in Light rain and Rain weathers are small. However, we argue that adverse weathers have impacts on the traffic flow volume and the results in our paper are based on the volume in clear sky while results in previous works are based on a lower volume (field data-based research), resulting in the differences. Whereas, Table 4 can prove that the combination of driving simulator and traffic simulation is usefully in the effect assessment of weathers on traffic flow.”

Table 4  Comparison between results in our paper and results in previous paper

Weather

Literature

Changes in capacity

Changes on Speed

Result in our paper

Results in literature

Result in our paper

Results in literature

Light rain

Rakha, Farzaneh et al. (2008)

-15.3%

-10~-11%

-3.1%

-8~-10%

Rain

Agarwal, Maze, et al. (2005)

-3.2%

-7~-8%

-2.0%

-8~-12%

Heavy rain

Smith, Byrne, et al. (2004)

-11.1%

-4~-10%

-7.6%

-5.0~-6.5%

Agarwal, Maze, et al. (2005)

-10~-17%

-4~-7%

Snow

Roh, Sharma, et al. (2014)

-43.7%

-25%

-19.2%

-

Heavy snow

Smith, Byrne, et al. (2004)

-25~-30%

-

Agarwal, Maze, et al. (2005)

-19~-27%

-11~-15%

“In the calibration part, the target is to let the average speed of simulation are the same as field data. This goal is not achieved in this research. It might because when calibrating VISSIM, only traffic volume is used and no other traffic characteristics are available in our research. Although we have worked hard to calibrate the VISSIM map, there still some differences in Figure 6. However, a consistent tendency is obtained. In terms of the objective of this paper that is measuring the effects of various adverse weathers on traffic flow characteristics, the relative changes could be convinced. Meanwhile, the trends in this research are in accordance with previous outcomes, which can also prove the relative effectiveness of the method that it can be used to measure the influence of weather on traffic flow. In future works, if the free flow scenario can be designed more carefully or if richer field data can be obtained to support a better base map drawing and a better calibration, the results would be more accurate.”

Question 8. The paper needs a proofreading. There are lots of grammar error and typos.

Response: Thank you for this comment. We have tried our best to improve our English writing.

Road

type

Weather

condition

CC0

(m)

CC1

(s)

CC2

(m)

CC3

(s)

CC4

(m/s2)

CC5

(m/s2)

CC6

(-)

CC7

(m/s2)

CC8

(m/s2)

CC9

(m/s2)

Ave±Std

Ave±Std

Ave±Std

Ave±Std

Ave±Std

Ave±Std

Ave±Std

Ave±Std

Ave±Std

Ave±Std

Basic segment

CS

4.54±   2.38

0.87±0.86

5.28±7.72

-7.92± 5.36

-1.52±0.3

1.52±0.3

0.71

0.31±0.2

1.03± 0.6

0.33± 0.16

LF

1.84± 0.81

1.51±1.33

6.38±5.17

-8.57± 6.52

-1.26±0.41

1.26±0.41

0.73

0.32±0.22

1.3± 0.56

0.3± 0.11

F

3.02± 4.5

1.44±1.05

7.48±11.82

-6.99± 7.64

-0.92±0.4

0.92±0.4

0.73

0.32±0.22

1.37± 0.62

0.32± 0.12

DF

1.78± 0.37

1.52±1.39

7.33±10.04

-7.47± 5.81

-0.83±0.42

0.83±0.42

0.64

0.33±0.19

1.28± 0.62

0.33± 0.19

HDF

1.6±5.65

1.26±9.09

19.4±31.61

-5.37± 13.31

-0.83±0.53

0.83±0.53

0.25

0.34±0.19

1.18± 0.55

0.26± 0.13

LR

9.54± 8.21

1.21±1.32

6.36±4.87

-7.47± 8.41

-0.67±0.33

0.67±0.33

0.64

0.34±0.22

1.3± 0.68

0.35± 0.16

R

1.06± 0.92

1.67±1.8

9.67±6.57

-7.24± 5.19

-0.64±0.28

0.64±0.28

0.56

0.35±0.19

1.34± 0.61

0.38± 0.15

HR

5.36± 1.98

1.45±1.32

10.7±11.42

-6.46± 9.67

-0.61±0.41

0.61±0.41

0.65

0.35±0.18

1.32± 0.69

0.33± 0.15

EHR

1.34± 0.95

2.31±6.2

11.2±26.48

-8.39± 6.93

-0.6±0.43

0.6±0.43

0.68

0.36±0.21

1.34± 0.71

0.31± 0.09

S

2.33± 2.96

3.93±1.28

16±9.91

-7.01± 6.06

-0.59±0.28

0.59±0.28

0.64

0.38±0.21

1.37± 0.67

0.32± 0.15

EHS

1± 0.67

10.88±10.33

20±44.32

-8.09± 9.58

-1.43±0.46

1.43±0.46

0.57

0.39±0.15

1.36± 0.68

0.3± 0.13

Upslope

CS

4.54± 2.38

1.3±1.37

8.58±9.95

-7.92± 5.36

-2.1±0.47

2.1±0.47

0.62

0.4±0.22

1.03± 0.6

0.33± 0.16

LF

1.84± 0.81

1.28±1.28

8.89±6.96

-8.57± 6.52

-2.36±0.43

2.36±0.43

0.60

0.37±0.24

1.3± 0.56

0.3± 0.11

F

3.02± 4.5

1.25±1.06

11.68±7.13

-6.99± 7.64

-0.9±0.43

0.9±0.43

0.68

0.38±0.22

1.37± 0.62

0.32± 0.12

DF

1.78± 0.37

1.7±1.07

3.73±10.96

-7.47± 5.81

-0.98±0.5

0.98±0.5

0.75

0.36±0.2

1.28± 0.62

0.33± 0.19

HDF

1.6± 5.65

1.18±3.86

20.7±9.99

-5.37± 3.31

-1.15±0.76

1.15±0.76

0.50

0.36±0.23

1.18± 0.55

0.26± 0.13

LR

9.54± 8.21

1.08±1.23

13.33±6.66

-7.47± 8.41

-1.41±0.44

1.41±0.44

0.68

0.38±0.21

1.3± 0.68

0.35± 0.16

R

1.06± 0.92

1.27±1.13

18.02±11.51

-7.24± 5.19

-0.93±0.47

0.93±0.47

0.69

0.38±0.21

1.34± 0.61

0.38± 0.15

HR

5.36± 1.98

1.26±1.08

7.47±7.58

-6.46± 9.67

-0.87±0.5

0.87±0.5

0.68

0.37±0.21

1.32± 0.69

0.33± 0.15

EHR

1.34± 0.95

2.36±5.42

19.43±26.06

-8.39± 6.93

-1.02±0.49

1.02±0.49

0.57

0.36±0.21

1.34± 0.71

0.31± 0.09

S

2.33± 2.96

4.33±1.39

16±9.37

-7.01± 6.06

-0.79±0.57

0.79±0.57

0.68

0.37±0.21

1.37± 0.67

0.32± 0.15

EHS

1± 0.67

6.74±10.23

20±31.91

-8.09± 9.58

-1.06±0.73

1.06±0.73

0.47

0.4±0.25

1.36± 0.68

0.3± 0.13

Downslope

CS

4.54± 2.38

0.56±0.5

3.84±5.61

-7.92± 5.36

-2.53±0.34

2.53±0.34

0.63

0.42±0.22

1.03± 0.6

0.33± 0.16

LF

1.84± 0.81

0.82±0.63

4.64±2.34

-8.57± 6.52

-2.19±0.39

2.19±0.39

0.54

0.47±0.24

1.3± 0.56

0.3± 0.11

F

3.02± 4.5

0.73±0.75

6.99±8.78

-6.99± 7.64

-1.84±0.39

1.84±0.39

0.56

0.44±0.16

1.37± 0.62

0.32± 0.12

DF

1.78± 0.37

1.01±0.94

3.02±3.38

-7.47± 5.81

-1.82±0.37

1.82±0.37

0.52

0.38±0.17

1.28± 0.62

0.33± 0.19

HDF

1.6± 5.65

0.67±6.7

11.91±16.73

-5.37± 13.31

-1.83±0.62

1.83±0.62

0.42

0.41±0.25

1.18± 0.55

0.26± 0.13

LR

9.54± 18.21

0.57±0.79

6.31±3.47

-7.47± 8.41

-1.73±0.41

1.73±0.41

0.55

0.41±0.2

1.3± 0.68

0.35± 0.16

R

1.06± 0.92

0.67±1.18

16.28±6.3

-7.24± 5.19

-1.56±0.39

1.56±0.39

0.36

0.4±0.17

1.34± 0.61

0.38± 0.15

HR

5.36± 1.98

0.63±0.79

5.78±4.76

-6.46± 9.67

-1.63±0.44

1.63±0.44

0.51

0.42±0.18

1.32± 0.69

0.33± 0.15

EHR

1.34± 0.95

2±3.58

14.13±14.2

-8.39± 6.93

-1.76±0.38

1.76±0.38

0.47

0.45±0.21

1.34± 0.71

0.31± 0.09

S

2.33± 2.96

4.11±1.04

16±7.73

-7.01± 6.06

-1.68±0.39

1.68±0.39

0.52

0.43±0.26

1.37± 0.67

0.32± 0.15

EHS

1± 0.67

8.14±11.42

20±20.55

-8.09± 9.58

-1.48±0.59

1.48±0.59

0.38

0.46±0.19

1.36± 0.68

0.3± 0.13

Reviewer 2 Report

This study is really interesting and focuses on how weather affects the traffic flow characteristic using driving simulator and traffic simulator. The reviewer has following comments:

This paper mention the following meaning "assess the effects of weather on traffic flow by combining driving simulator and traffic simulation" in different words.  It is not necessary to mention 
throughout the paper.

Figure 1, car following model parameters input to VISSIM software. I think green arrow represents opposite direction. Why another arrow is going from car following model to driver behavior collection and what is the role of traffic demand? Really this figure is confusing, so redraw with explanation.

SetFog is used to control the visibility with the range from 0 m to 10000 m. 10000 m is too large. The driver won't be visible within 10 km. Modify the experiment based on real world field values. I think less than 800 m gives realistic experiment.

Line number 240-244 does not explain accelerating process, it is deceleration process.

In table 3, clear sky desired speed is lower than other fog condition. This is not logical. Generally, in adverse weather desired speed is less. Why this happens?

In table 3, Parameter cc0 (Standstill distance) term for  clear sky is higher than all other fog situations. Generally, in good weather conditions, the driver standstill distance is very less
compared  to adverse weather condition due to driver's increase visibility and perception. This is not logical. Check the whole table 3, is there logical inconsistency? or samples not enough? or experimental error?

Line number 344-345, the differences between field speed and simulation speed for each road section are about 15-20km/h. This is really huge difference 33% to 50%. 
Generally, within 20% error is the acceptable limit for macroscopic level calibration.

Author Response

Thank you for reviewing my manuscript and the comments. We have revised our manuscript according to your comments.

This study is really interesting and focuses on how weather affects the traffic flow characteristic using driving simulator and traffic simulator. The reviewer has following comments:

Question 1. This paper mention the following meaning "assess the effects of weather on traffic flow by combining driving simulator and traffic simulation" in different words. It is not necessary to mention throughout the paper.

Response: It’s my apologies that we have mentioned this sentence too many times. We have modified some of the sentences in my manuscript.

Question 2. Figure 1, car following model parameters input to VISSIM software. I think green arrow represents opposite direction. Why another arrow is going from car following model to driver behavior collection and what is the role of traffic demand? Really this figure is confusing, so redraw with explanation.

Response: Thank you for this comment. There do is an error in the green arrow. The green arrow should go from ‘VISSIM’ to ‘Car-following model’ and now we have corrected this error. The ‘Demand’ between ‘Driver behavior collection’ and ‘Car-following model’ is deleted to make this picture clearer. The original purpose of the ‘Demand’ is to convey this information: ‘The collection of driver behavior parameters is based on the demand of car-following model.’ We find that this information may not be so important and on the contrary will make this picture hard to understand. Thus, it is deleted. Figure 1 in our manuscript has been replaced and also shown in this letter:

Question 3. SetFog is used to control the visibility with the range from 0 m to 10000 m. 10000 m is too large. The driver won't be visible within 10 km. Modify the experiment based on real world field values. I think less than 800 m gives realistic experiment.

Response: Thank you for this comment. The 10000m is the maximum value that can be input in SetFog function due to the limitation of driving simulator software. 10000m is the default setting that represents the clear sky. Drivers won’t see that far distance. Thus, the setting of 10000 means fog has no effect on drivers, i.e. the clear sky. We have added some explanation in our manuscript on line 240-241:

“SetFog is used to control the visibility with the range from 0m to 10000m (10000m is the default setting that means clear sky, i.e. the fog has no effect on drivers’ visibility distance).”

Question 4. Line number 240-244 does not explain accelerating process, it is deceleration process.

Response: Thank you for this comment. This is a spelling error. It has been revised.

Question 5. In table 3, clear sky desired speed is lower than other fog condition. This is not logical. Generally, in adverse weather desired speed is less. Why this happens?

Response: Thank you for this comment. The desired speed is extracted in the free-flow scenario and average speed is used as the desired speed. We have analyzed the influence of weather condition on desired speed. It is found that except for the values of HDF, there is no significant difference between the values of other weathers. The slight difference between the desired speeds might be caused by the complex effect of the visibility distance and road surface friction. This phenomenon is different with our intuition, However, the desired speed is tested in idea environment (road length is 10km. Only a few cars is on the road). The results may be reasonable. The desired speed is input into VISSIM and functions as the max speed when the driving environment is available. Thus it may not have a huge effect on the results. We have added some comparison between the results in Figure 7 and results in previous research (See line 481-489) and this comparison shows that the results are rational. Thus, it can prove that the values of the desired speed are available in some way. In future works, a more carefully scenario design for free flow should be focused to eliminate this question. To address this issue, we have added some explanation in Discussion section: See line 505-515:

“Another one issue is the desired speed in Table 3. The desired speed of clear sky is lower than some of the desired speed of adverse weathers. We have analyzed the influence of weather condition on desired speed. It is found that except for the values of HDF, there is no significant difference between the values of other weathers. The slight difference between the desired speeds might be caused by the complex effect of the visibility distance and road surface friction. This phenomenon is different with our intuition, However, the desired speed is tested in idea environment (road length is 10km. Only a few cars is on the road). The results may be reasonable. The desired speed is input into VISSIM and functions as the max speed when the driving environment is available. Thus it may not have a huge effect on the results. In future works, a more carefully scenario design for free flow should be focused to eliminate this question.”

Question 6. In table 3, Parameter cc0 (Standstill distance) term for a clear sky is higher than all other fog situations. Generally, in good weather conditions, the driver standstill distance is very less compared to adverse weather condition due to driver's increase visibility and perception. This is not logical. Check the whole table 3, is there logical inconsistency? or samples not enough? or experimental error?

Response: Thank you for this comment. The values in Table 3 is extracted from the driving behavior data by conducting the driving simulation experiment. No modification is applied in these values. We have examined the values of CC0 for clear sky and find that the higher values than other weathers maybe because that the complex effect of visibility distance and friction decrease make the deceleration difficult, leading to small standstill distance. Changes in other values are also due to similar reasons. Some of the values may be difficult to understand. However, the results in Figure 7 is reasonable, which proves that the values in Table 3 are also reasonable. In the future works, a careful analysis of each parameter should be conducted to have better results.

As a supplement, we added some explanation in the Discussion section. See line 481-489, 496-504:

“Table 4 shows the comparison between the results in our paper and results in previous paper. It can be seen that the results based on the proposed method are similar to that in previous research except for that the speed reduction in Light rain and Rain weathers are small. However, we argue that adverse weathers have impacts on the traffic flow volume and the results in our paper are based on the volume in clear sky while results in previous works are based on a lower volume (field data-based research), resulting in the differences. Whereas, Table 4 can prove that the combination of driving simulator and traffic simulation is usefully in the effect assessment of weathers on traffic flow.”

Table 4  Comparison between results in our paper and results in previous paper

Weather

Literature

Changes in capacity

Changes on Speed

Result in our paper

Results in literature

Result in our paper

Results in literature

Light rain

Rakha, Farzaneh et al. (2008)

-15.3%

-10~-11%

-3.1%

-8~-10%

Rain

Agarwal, Maze, et al. (2005)

-3.2%

-7~-8%

-2.0%

-8~-12%

Heavy rain

Smith, Byrne, et al. (2004)

-11.1%

-4~-10%

-7.6%

-5.0~-6.5%

Agarwal, Maze, et al. (2005)

-10~-17%

-4~-7%

Snow

Roh, Sharma, et al. (2014)

-43.7%

-25%

-19.2%

-

Heavy snow

Smith, Byrne, et al. (2004)

-25~-30%

-

Agarwal, Maze, et al. (2005)

-19~-27%

-11~-15%

“Some values in Table 3 seem to be odd such as the high CC0 value for clear sky. It should be noted firstly that the values in Table 3 is extracted from the driving behavior data from the driving simulation experiment and no modification is applied. The lower values of CC0 for adverse weathers may be caused by the complex effect of decrease of visibility distance and surface friction. Changes of other values may be also due to similar reasons. However, although there might some puzzle in these parameter values, the rationality of results in Figure 7 can proves that the values in Table 3 is also reasonable. And the proposed method in this paper can also be seen available.”

Question 7. Line number 344-345, the differences between field speed and simulation speed for each road section are about 15-20km/h. This is really huge difference 33% to 50%. Generally, within 20% error is the acceptable limit for macroscopic level calibration.

Response: Thank you for this comment. There do have some difference between the field data and simulation data. It might because when calibrating VISSIM, only traffic volume is used and no other traffic characteristics are available in our research.  But the focus of this paper is to emphasize the combination of driving simulator and traffic simulation. Compared with results in previous research, the results in Figure 7 seem to be rational. Thus, it is suggested that the proposed method is valid. The differences between field data and simulation data in Figure 6 can be eliminated by a more reasonable driving simulation experiment design and full calibration of traffic simulation if field data can be abundant. We have added some explanation on this issue, See line 516-527.

“In the calibration part, the target is to let the average speed of simulation are the same as field data. This goal is not achieved in this research. It might because when calibrating VISSIM, only traffic volume is used and no other traffic characteristics are available in our research. Although we have worked hard to calibrate the VISSIM map, there still some differences in Figure 6. However, a consistent tendency is obtained. In terms of the objective of this paper that is measuring the effects of various adverse weathers on traffic flow characteristics, the relative changes could be convinced. Meanwhile, the trends in this research are in accordance with previous outcomes, which can also prove the relative effectiveness of the method that it can be used to measure the influence of weather on traffic flow. In future works, if the free flow scenario can be designed more carefully or if richer field data can be obtained to support a better base map drawing and a better calibration, the results would be more accurate.”

Reviewer 3 Report

The paper addresses an interesting and potentially practitioners-useful research question. The method is not innovative nor convincing. The results are minimal and not generalizable. The discussion is poor. The contribution to the research literature is lacking.

·         The motivation and scope of the study should be better and extensively specified. Please, indicate which gaps in the current literatures you want to address in your research, if any.

·         Both the research tools used in the study need to be validated. The issue of validation is not approached in the whole paper. Specifically, a driving simulator can be an effective tool for addressing driving behavior issues, but it needs to be correctly validated for the specific application. No reference to validation of driving simulator of Beijing University of Technology is reported. This point is fundamental for the transferability of study results and conclusions from the simulated to the real environment.

·          “Combination of driving simulator and traffic simulation” section: The description of driving simulator is very poor. Given the importance of the simulator set up to our interpretation of the question of fidelity and validity, it would be better to have any and all pertinent information: characteristics of simulator controls (gearbox type, steering wheel force-feedback, etc.) and of motion platform (if any) could effectively impacts on simulator driver behaviour (in particular, longitudinal acceleration during car-following). What’s the used driving simulator software? And the vehicle dynamic software (friction simulation)?

·         “Driving simulation experiment and parameter extraction” section: It’s not clear as the surroundings traffic in driving simulator environment is simulated. Is commercial microsimulation software used or is there a specific traffic module of the driving simulator software? How the three traffic flow states have been simulated? And which the relative density and traffic flow values are?

·         “Scenario design” section: A more detailed description of the simulated road has to be added, such as characteristics of longitudinal grade, shoulders, markings, signs, and surrounding environment.

·         “Driving simulation experiment and parameter extraction” section: Drivers are expressly required by experimenter to carry out a car-following manoeuvre or the manoeuvre develops naturally, and the driver is free to perform lanes change or overtaking. the interrupted car-following manoeuvres were discarded or were used in the extraction of the parameters for the traffic simulator.

·         How many car-following manoeuvres in total are carried out by each driver?

·         “Weather design” section: The use of relative measures for meteorological/climatic quantities makes particularly difficult to reproduce the scenarios in future studies, to transfer the results in reality and to compare them with other studies. The same applies to friction.

·         “Weather design” section: During rain or snow, has the wiper functioning been simulated? Have the fog lights (rear and/or front) been simulated (In Europe, the rear fog lights are mandatory in dense rain, fog, dust or falling snow)?

·         “Experiment implementation” section: The sample is clearly unbalanced respect to gender. The choice to select only professional drivers is a further limitation and has a significant impact on generalizability of the results.

·         “Experiment implementation” section: It seems that no rest was previewed between each scenario, is it true? Did the subjects perform a familiarization drive before the test sessions?

·         In absence of information, I suppose that no participant dropped out of the experiment because of simulator sickness. Is the simulator sickness monitored? What about the used metrics of simulator discomfort? What were discomfort scores of the participants and how did these affect results? A slight discomfort is not obvious or easy to detect and could alter the drivers behavior (reducing speed and acceleration, avoiding abrupt steering maneuverers and so on) invalidating the results.

·          “Calibrating of traffic flow distribution” sub-section: Calibration results are not particularly convincing. The differences between average field speeds and average simulation speeds, during light rain (Figure 6c and 6d), cannot be neglected ranging from 25 to 35 Km/h (the simulation speeds are more than double that on- road speeds). This issue and its impact on study results should be better and extensively addressed in the discussion.

·         The discussion is weak. It is expected that the authors provide their interpretation, explain the implications of the findings and make an explicit and exhaustive comparison with the existing literature. I would urge to authors to consider a more integrative approach here. Study limitations needs to be addressed also in the Discussion Section

·         An adequate discussion on the limitation in the use of driving simulator such as realism, simulator sickness, simulator validation, driver motivation, and level of perceived risk in a simulated environment and its impacts on driver’s performance should be properly added in the paper. Also, the possible limitations to the study due to the sample characteristics (unbalanced gender distribution, 100% of professional drivers) need to be addressed in the Discussion Section.

·         “Conclusion” section: Considering that there is only a sketch of relative validity (still to be verified) in the results it would be advisable not to declare precise values of changes in average speed and road capacity.

·         Reading the paper, it’s not evident any significant and conclusive contribution to the research literature. How does this paper add to the literature on traffic flow characteristics during adverse weather?

·         Line 240: “decelerating car-following” instead of “accelerating car-following”.

·         Table 3: Something sounds wrong in this table: the values of parameters extract from driving simulation experiments are exactly the same in each of the three road type considered.

·         Figure 7: To increase the readability of the figure, it is advisable to use a different polyline for each weather condition tested (CS, Fog, Rain, Snow).

·         Lines 527-528: some information on the reference is lacking. Please check all the references and use the formatting guidelines of the journal for the “References” section and in-text citations.

Author Response

Reply to reviewer #3:

Thank you for reviewing my manuscript and the comments. We have revised our manuscript according to your comments.

The paper addresses an interesting and potentially practitioners-useful research question. The method is not innovative nor convincing. The results are minimal and not generalizable. The discussion is poor. The contribution to the research literature is lacking.

Response: Thank you for your comment. We have tried our best to revise this paper.

Question 1. The motivation and scope of the study should be better and extensively specified. Please, indicate which gaps in the current literatures you want to address in your research, if any.

Response: Thank you for this comment. The aim of this study is to propose a method that is used to assess the influence of adverse weathers on traffic flow. The motivation why we conduct this research is to overcome the shortcoming of field data-based research, i.e. adverse weather data and the traffic volume is unable to control. By conducting valid experiments, the results show that this proposed method is useful and the results are in accordance with the results in previous works. To better explain our aim and motivation, we have modified the last paragraph in Introduction section. See line 151-154:

“To summary, an in-deep and comprehensive analysis of the effect of adverse weathers on traffic flow is essential to help to the countermeasures development. However, field data-based studies are limited by the uncontrollable of adverse weathers, making the accurate analysis difficult. This paper aims to propose a method of using the driving simulator and traffic simulation to assess the influence of weather conditions on traffic flow, overcoming the limitations of the actual data-based approach. Both the research on the impacts of weather and the effects of other factors on traffic flow can be studied using this proposed approach to have a deep analysis from the angle of driver behaviors.”

Question 2. Both the research tools used in the study need to be validated. The issue of validation is not approached in the whole paper. Specifically, a driving simulator can be an effective tool for addressing driving behavior issues, but it needs to be correctly validated for the specific application. No reference to validation of driving simulator of Beijing University of Technology is reported. This point is fundamental for the transferability of study results and conclusions from the simulated to the real environment.

Response: Thank you for this comment. I apologize that we did not explain the information on driving simulator in our research. This driving simulator has supported multiple studies related to driving behavior. In previous researches, the application in driving behavior related studies has also been proved valuable. As a supplement, we have added the introduction in Line 202-218:

“Apparatus

This research used a fixed-base driving simulator located at Beijing University of Technology, imported from Korea, produced by INNO-Simulation Company. The simulator includes a modified car (replacing the original vehicle accessories with computers or dynamic sensors), control computers and video and audio devices. Driving circumstances are projected onto four large screens (three ahead of and one behind the simulator car) and are displayed on two small screens on both sides of the car as side mirrors. This driving simulator is controlled by an embedded software called SCANeR Studio that is also developed by the producer. Using this software, the driving scenario can be fully controlled to perform nearly-true driving experience. Besides, the software record driver behavior (e.g., gas pedal, brake pedal, steering wheel angle) and vehicle operation data (e.g., speed, acceleration, distance to lead/rear car, X/Y coordinates) during experiments at 1-50 Hz.

The validity of this simulator in studying driving behavior has been verified in previous research (Ding, Zhao, et. al., Zhao, Guan, et. al.). A total of 250 drivers have participated in driving-behavior-related research on this driving simulator, and this simulator has also been evaluated through questionnaires. The average score of the reality of the driving simulator reaches 8 (1-not real at all to 10-very real). Thus, research based on this simulator is considered valuable.”

Question 3. Combination of driving simulator and traffic simulation” section: The description of driving simulator is very poor. Given the importance of the simulator set up to our interpretation of the question of fidelity and validity, it would be better to have any and all pertinent information: characteristics of simulator controls (gearbox type, steering wheel force-feedback, etc.) and of motion platform (if any) could effectively impacts on simulator driver behaviour (in particular, longitudinal acceleration during car-following). What’s the used driving simulator software? And the vehicle dynamic software (friction simulation)?

Response: Thank you for this comment. The used driving simulator is imported from Korea and produced by a company that is professional in the field of driving simulation. This driving simulator uses nine computers (1 main PC, 1 Dynamic controller PC, 5 visual PC, 1 Motion controller PC, and 1 data collection PC), four projectors (3 for the head screen and 1 for the rear screen), two small screens (side mirrors). The manipulate information like steering wheel, gas pedal, gearbox is recorded by electric sensors. The control software is also developed by the producer too. Due to that this company is professional in the development of driving simulator, it is thought that vehicle dynamic characteristics (controlled by the software) are rational. The introduction of the used driving simulator has been added in Line 202-218.

“Apparatus

This research used a fixed-base driving simulator located at Beijing University of Technology, imported from Korea, produced by INNO-Simulation Company. The simulator includes a modified car (replacing the original vehicle accessories with computers or dynamic sensors), control computers and video and audio devices. Driving circumstances are projected onto four large screens (three ahead of and one behind the simulator car) and are displayed on two small screens on both sides of the car as side mirrors. This driving simulator is controlled by an embedded software called SCANeR Studio that is also developed by the producer. Using this software, the driving scenario can be fully controlled to perform nearly-true driving experience. Besides, the software record driver behavior (e.g., gas pedal, brake pedal, steering wheel angle) and vehicle operation data (e.g., speed, acceleration, distance to lead/rear car, X/Y coordinates) during experiments at 1-50 Hz.

The validity of this simulator in studying driving behavior has been verified in previous research (Ding, Zhao, et. al., Zhao, Guan, et. al.). A total of 250 drivers have participated in driving-behavior-related research on this driving simulator, and this simulator has also been evaluated through questionnaires. The average score of the reality of the driving simulator reaches 8 (1-not real at all to 10-very real). Thus, research based on this simulator is considered valuable.”

Question 3. “Driving simulation experiment and parameter extraction” section: It’s not clear as the surroundings traffic in driving simulator environment is simulated. Is commercial microsimulation software used or is there a specific traffic module of the driving simulator software? How the three traffic flow states have been simulated? And which the relative density and traffic flow values are?

Response: Thank you for this comment. The surrounding traffic in the driving simulation is simulated using the driving simulator software. The software provides a function that can control the speed of surrounding vehicles. The three traffic flow stat is simulated by adjusting the speed of the surrounding vehicle to 40, 70 and 80km/h. In the first two scenarios, there are about 27 surrounding vehicles. The difference is on the speed, which in turn resulting in different traffic flow states. The headway time for the surrounding vehicle is set to 2~3.5 randomly. According to the speed and headway time, the relative density of low-speed scenario and high-speed scenario are 26~45 pcu/ln/km and 15~25pcu/ln/km respectively. And the relative traffic flow rate is 1028~1800 pcu/ln/h. We have added detail explanation on the design of traffic flow state. See Line 269-272:

“The variation of traffic flow state in the driving simulator is configured by setting the speeds of surrounding vehicles, which is one of the functions of driving simulator software. Safety headway time of each surrounding vehicle is set as 2~3.5s randomly. In this case, the combined configuration of speed and headway time can produce different traffic flow state.”

Question 4. “Scenario design” section: A more detailed description of the simulated road has to be added, such as characteristics of longitudinal grade, shoulders, markings, signs, and surrounding environment.

Response: Thank you for this comment. The scenario is built referring to the actual road in Beijing. The number of lanes (mostly is 3), lane length (mostly is 4m), the position of exit and entrance, the structure of bridges (longitudinal grade is 1.5), and markings and signs in the simulator is nearly the same as the actual road. Due to the huge workload in the drawing of scenario, some road parameters like the width of shoulders use the default settings in China’s road design standard. We have added some explanation on the design of scenario in line 224-225:

“Other road parameter like the position of exit and entrance, the structure of bridges, and markings and signs in the simulator is also designed according to the actual road.”

Question 5. “Driving simulation experiment and parameter extraction” section: Drivers are expressly required by experimenter to carry out a car-following manoeuvre or the manoeuvre develops naturally, and the driver is free to perform lanes change or overtaking. the interrupted car-following manoeuvres were discarded or were used in the extraction of the parameters for the traffic simulator.

Response: Thank you for this comment. Drivers are required to perform car-following maneuver and lane-change or overtaking are forbidden. If a car-following progress is interrupted (by a crash or lane-change), the corresponding data will be discarded in the extract of car-following parameter. We have added this explanation in line 306-308:

“If car-following progress is interrupted (by a crash or lane-change), the corresponding data will be discarded in the extract of the car-following parameter.”

Question 6. How many car-following manoeuvres in total are carried out by each driver?

Response: Thank you for this comment. Drivers are required to follow the lead car all the time.

Question 7. Weather design” section: The use of relative measures for meteorological/climatic quantities makes particularly difficult to reproduce the scenarios in future studies, to transfer the results in reality and to compare them with other studies. The same applies to friction.

Response: Thank you for this comment. The driving simulator used in this research is imported from Korea and the functions that control rain and snow is embedded in its software. The unit of these two function is %. There surely have some problems but we cannot change this setting. To solve this problem, before the experiment, we conducted a matching experiment to match the weather scenario in a driving simulator and the actual weathers. A questionnaire is used in this matching experiment and total 30 students participated in this matching experiment. The main task of this questionnaire for these participants is to link the weather scenario in driving simulator with the actual weather conditions and label the similarity (%). Grades of actual weather conditions (China unit) are also provided in the questionnaire. According to the results of the 30 participants’ questionnaires, we matched the simulated scenario with actual weathers. The information of the actual weather matching the simulated scenario is now added in Table 1. See line 249-259:

“To match the simulated weather conditions to actual weather grades, a 30-person-participated visual experiment that aims at establishing the corresponding relationship between the driving simulation weather environment and actual weather grade that used in the weather grading in China. Students chose the most similar actual weather grade for every simulated weather condition after the observation of simulated weathers. The matching results are shown in the column “Matched weather condition” in Table 1 as well. The visual effects of some of the 11 weather conditions are shown in Figure 3.”

TABLE 1  Configuration of 11 weather conditions

ID

Configuration of   Three Functions

Matched Weather Condition

(actual weather grade)

Abbreviation

SetRain (%)

SetSnow (%)

SetFog (m)

Friction (%)

1

-

-

10000

100

Clear   Sky

CS

2

-

-

1500

100

Light Fog

(1000S100,000)

LF

3

-

-

800

100

Fog

(500S1000)

F

4

-

-

300

100

Dense Fog

(200S500)

DF

5

-

-

50

100

Heavy Dense Fog

(50S200)

HDF

6

20

-

2000

100

Light Rain

(0-9.9)mm/24h

LR

7

45

-

800

75

Rain

(10.0-24.9)mm/24h

R

8

70

-

550

60

Heavy Rain

(25.0-49.9)mm/24h

HR

9

95

-

300

45

Extremely Heavy Rain

(100.0-249.0)mm/24h

EHR

10

-

45

500

45

Snow

(2.5-4.9)mm/24h

S

11

-

95

100

20

Extremely Heavy Snow

(10-19.9)mm/24h

EHS

Similarly, the road surface friction setting is limited by the driving simulator software and it will change as the change of the value of SetFog and SetRain. Due to this company is professional in the development of driving simulator, it is thought that the setting of friction is rational.

Question 7. Weather design” section: During rain or snow, has the wiper functioning been simulated? Have the fog lights (rear and/or front) been simulated (In Europe, the rear fog lights are mandatory in dense rain, fog, dust or falling snow)?

Response: Thank you for this comment. During our experiment, the wiper didn’t work because that our simulator cannot simulator the visual effects of ‘rain on the glass’ or ‘snow on the glass’. The rear fog lights of the surrounding vehicle are opened in when visibility distance is low (<100m, automatically controlled by the driving simulator software). The participants are not allowed to use headlights when driving because that opened headlight will slightly increase the visibility distance, which may influence the desired setting of scenario. We have added this explanation on Line 332-334:

“During the driving experiment, participants are not allowed to use headlights when driving because that opened headlight will slightly increase the visibility distance, which may influence the desired setting of scenario.”

Question 8. “Experiment implementation” section: The sample is clearly unbalanced respect to gender. The choice to select only professional drivers is a further limitation and has a significant impact on generalizability of the results.

Response: Thank you for this comment. The selection of 7 female drivers is based on the drivers’ demographic characteristics in China where female drivers take 25 percent in all of the drivers. The selection of professional drivers is really a limitation. In this research, the aim is to propose and verify the combination of driving simulator and traffic simulation. The using of professional drivers can be seen as a case study. In future works or further application, various types of drivers need to be taken into consideration to have better results. See line 490-492:

In this paper, participants in driving simulation experiment are professional drivers, which may lead to bias on the results. In future works or further application, various types of drivers need to be taken into consideration to have better results.

Question 9. “Experiment implementation” section: It seems that no rest was previewed between each scenario, is it true? Did the subjects perform a familiarization drive before the test sessions?

Response: Thank you for this comment. I apologize that we didn’t explain the process of experiment implementation in detail. We have added in this section. See line 318-324, 328-320:

“Once a driver arrives at the driving simulation laboratory, a pre-driving tutorial that lasts for approximately 10 minutes must be performed to help the driver adapt to the experiment equipment. Then, the participant was asked to complete a questionnaire about the participant’s basic information, including age, age of driving, time to fall asleep, time to wake up, and whether on medication or drunk. If a driver slept less than 6 hours or took medicine or a drink during the past 12 hours, his/her experiment would be canceled and adjusted to a later time.”

“The average driving time for each scenario is about 7~10 minutes, followed by a rest lasts for about 3 min when the driving simulator loading next scenario”

Question 10. In absence of information, I suppose that no participant dropped out of the experiment because of simulator sickness. Is the simulator sickness monitored? What about the used metrics of simulator discomfort? What were discomfort scores of the participants and how did these affect results? A slight discomfort is not obvious or easy to detect and could alter the drivers behavior (reducing speed and acceleration, avoiding abrupt steering maneuverers and so on) invalidating the results.

Response: Thank you for this comment. Simulator sickness is always a problem in the experiments based on a driving simulator. Due to we have conducted several studies based on our driving simulator, we have accumulated a driver database. Drivers in this database can drive the simulated vehicle without discomfort. That is to say that no drivers give up experiment because of the simulator sickness. See line 492-495:

“These participants is chosen from a driver database that is established by previous driving simulation studies. Thus these drivers don’t have simulator sickness and can correctly drive, and the driving behavior is suggested to be available in the analysis of car-following driving behaviors.”

Question 11. Calibrating of traffic flow distribution” sub-section: Calibration results are not particularly convincing. The differences between average field speeds and average simulation speeds, during light rain (Figure 6c and 6d), cannot be neglected ranging from 25 to 35 Km/h (the simulation speeds are more than double that on- road speeds). This issue and its impact on study results should be better and extensively addressed in the discussion.

Response: Thank you for this comment. There do have some difference between the field data and simulation data. But the focus of this paper is to emphasize the combination of driving simulator and traffic simulation. Compared with results in previous research, the results in Figure 7 seem to be rational. Thus, it is suggested that the proposed method is valid. The differences between field data and simulation data in Figure 6 can be eliminated by a more reasonable driving simulation experiment design and full calibration of traffic simulation if field data can be abundant. We have added some explanation on this issue in Discussion section, See line 516-527:

“In the calibration part, the target is to let the average speed of simulation are the same as field data. This goal is not achieved in this research. It might because when calibrating VISSIM, only traffic volume is used and no other traffic characteristics are available in our research. Although we have worked hard to calibrate the VISSIM map, there still some differences in Figure 6. However, a consistent tendency is obtained. In terms of the objective of this paper that is measuring the effects of various adverse weathers on traffic flow characteristics, the relative changes could be convinced. Meanwhile, the trends in this research are in accordance with previous outcomes, which can also prove the relative effectiveness of the method that it can be used to measure the influence of weather on traffic flow. In future works, if the free flow scenario can be designed more carefully or if richer field data can be obtained to support a better base map drawing and a better calibration, the results would be more accurate.”

Question 12. The discussion is weak. It is expected that the authors provide their interpretation, explain the implications of the findings and make an explicit and exhaustive comparison with the existing literature. I would urge to authors to consider a more integrative approach here. Study limitations needs to be addressed also in the Discussion Section

Response: Thank you for this comment. We have added some discussion in that section according to the comments. The added contents mainly include a description of the results, the explanation on the values of Table 3, the limitation of the research etc.

Question 13. An adequate discussion on the limitation in the use of driving simulator such as realism, simulator sickness, simulator validation, driver motivation, and level of perceived risk in a simulated environment and its impacts on driver’s performance should be properly added in the paper. Also, the possible limitations to the study due to the sample characteristics (unbalanced gender distribution, 100% of professional drivers) need to be addressed in the Discussion Section.

Response: Thank you for this comment. We have added some discussion in that section according to the comments. The added contents mainly include a description of the results, the explanation on the values of Table 3, the limitation of the research etc.

Question 14. “Conclusion” section: Considering that there is only a sketch of relative validity (still to be verified) in the results it would be advisable not to declare precise values of changes in average speed and road capacity.

Response: Thank you for this comment. The conclusion has been modified and the precise values of changes in traffic flow have been deleted.

Question 15. Reading the paper, it’s not evident any significant and conclusive contribution to the research literature. How does this paper add to the literature on traffic flow characteristics during adverse weather?

Response: Thank you for this comment. The aim of this paper is to propose a method of using the driving simulator and traffic simulation to assess the influence of weather conditions on traffic flow, Although there are some shortcoming in this research like the imperfect calibration, the results about the reduction of traffic capacity and speed in our research are similar with the results in previous studies. Thus, the proposed method is proved to be available. This method can make up for the shortcomings of field data-based research and provided a new perspective for the understanding of the mechanisms why traffic flow changes. See Page 27, Line 547-550.

Question 16. Line 240: “decelerating car-following” instead of “accelerating car-following”.

Response: Thank you for this comment. This is a spelling error. It has been revised.

Question 17. Table 3: Something sounds wrong in this table: the values of parameters extract from driving simulation experiments are exactly the same in each of the three road type considered.

Response: Thank you for this comment. Parameters that have same values in each of the three road type include CC0 (Standstill distance), CC3 (Threshold for entering Following), CC8 (Standstill acceleration), CC9 (Acceleration with 80km/h) and desired speed. It can be easily understand that these parameters are slightly influenced by road type. For example, the CC0 is the distance when stop, which is thought to have nothing to do with slopes. The detail explanation is shown in line 345-352.

Question 18. Figure 7: To increase the readability of the figure, it is advisable to use a different polyline for each weather condition tested (CS, Fog, Rain, Snow).

Response: Thank you for this comment. Figure 7 has been modified.

Question 19. Lines 527-528: some information on the reference is lacking. Please check all the references and use the formatting guidelines of the journal for the “References” section and in-text citations.

Response: Thank you for this comment. We have examined this reference and it was found that this reference is student research that published as a library research award. So there is no other information about this reference. You can find this reference on Page: [http://broncoscholar.library.cpp.edu/handle/10211.3/193160]. The format of all of the reference has been modified according to the guidelines of the journal.

Round 2

Reviewer 1 Report

The authors have addressed all my comments.

Author Response

We thank you for the time you have spent reviewing and improving our manuscript.

Reviewer 3 Report

I appreciate the effort of authors to improve the paper. Most of my questions has been adequately addressed. Moreover, some points need further attention.

·         Lines 200-204:  As reported in these references (Ding et al., 2013; Zhao et al, 2013), your driving simulator (ds) has not undergone any behavioral validation procedure (behavior parameters on road VS behavior parameter on ds). The questionnaires on realism of ds concern more the fidelity than the validity of the simulator. This information must be clear and available to readers.

·         Lines 252 -255: Reading these lines and your response to my third question, it’s seems that you have used the “swarm” function of the SCANeR Studio, is it true? Unfortunately, this function is developed to guarantee a bubble of surrounding vehicles that constantly moves around the human driver, without considering any traffic law. Setting speed and time headway, does not guarantee specific traffic flows when “swarm” function is activated.

·         Lines 286-288: A driver involved in a crash at the ds should immediately stop driving and his participation in the experiment. This is because his driving behaviour could very likely be altered: fear, stress, excessive caution, unrealistic sensations, etc can normally occur.

Maybe is better: “If car-following progress is interrupted (by lane-change or overtaking), the corresponding data will be discarded in the extract of the car-following parameter.” instead of “If car-following progress is interrupted (by a crash or lane-change), the corresponding data will be discarded in the extract of the car-following parameter.”.

·         Response to my sixth question: Please, provide the number of effective car-following manoeuvres carried out by each driver.

·         Line 295: based on the gender composition of drivers” is not really clear. “based on the demographic characteristics of Chinese drivers” sounds better. Please provide a reference for this last statement.

·         Lines 459-462: The use of a driver database from which drawn your driver samples is only a partial solution to the issue of the driving simulator sickness because it depends by various factors, such as characteristics of driving scenario and psychophysics conditions of drivers. So, the probability that the same driver/subject became sick in a driving simulator varies over time and by type of experiments (length, complexity, driving environment, etc). Moreover, it’s mandatory to always monitor the level of simulator discomfort of each participants to avoid after-effects in the results. Therefore, I strongly recommend the authors to monitor simulator sickness in the future experiments.

Given the various limitations of experimental protocol, the partially convincing methodology and modest results, the article continues to be a little limited and not totally conclusive. However, a contribution, although minimal, to research literature is identifiable.

Ding, H., Zhao, X., Rong, J., and Ma, J. Experimental research on the effectiveness of speed reduction markings based on driving simulation: A case study. Accident Analysis & Prevention, 2013, 60, 211-218. 

Zhao X, Guan W and Liu X. A pilot study verifying how the curve information impacts on the driver performance with cognition model. Discrete Dynamics in Nature and Society, 2013, Article ID 316896, 1-8 .

Author Response

Dear editors and reviewers:

We thank you for the time you have spent reviewing and improving our manuscript entitled “Assessing the Influence of Adverse Weather on Traffic Flow Characteristics using Driving Simulator and VISSIM”. All of the suggestions comments provided by the reviewers have been addressed in the revised manuscript, and detailed responses to those comments are attached in this letter.

Sincerely,

Xiaohua Zhao Ph.D., Corresponding Author, Professor

The Beijing Engineering Research Center of Urban Transport Operation Guarantee

The College of Metropolitan Transportation, Beijing University of Technology

Beijing 100124, P.R. China, Tel.: +86-10-6739-6075; Email: [email protected]

Reply to reviewer #3:

Thank you for reviewing my manuscript and the comments. We have revised our manuscript according to your comments.

I appreciate the effort of authors to improve the paper. Most of my questions has been adequately addressed. Moreover, some points need further attention

Question 1. Lines 200-204: As reported in these references (Ding et al., 2013; Zhao et al, 2013), your driving simulator (ds) has not undergone any behavioral validation procedure (behavior parameters on road VS behavior parameter on ds). The questionnaires on realism of ds concern more the fidelity than the validity of the simulator. This information must be clear and available to readers.

Response: Thank you for this comment. We also conducted another one validation experiment to verify our driving simulator (New reference [28]). In the validation experiment, the subjects’ electroencephalogram and electrocardiogram of field and simulator were compared and results showed that driving simulation is absolutely effectiveness in straight sections and large radius corners sections. Although driving behavior parameters are not directly used in the validation experiment, it is thought that the relative effectiveness (comparison between treated groups and control group) of our driving simulation is valid in analyzing driving behaviors.

We have added the new reference and some explanation in the Discussion section. See Line 484-493.

“One of the issues of driving simulator based studies is the validation of the driving simulator. To solve this problem, we have conducted some validation experiments [26-28] and require each of participant to complete a questionnaire used to evaluate the reality of our driving simulation. It is showed that results from our driving simulator are in accordance with that from filed test; and participants feel the driving experiences in our driving simulator is similar to the real world. Although driving behavior parameters are not directly used in the validation experiment, it is thought that the relative effectiveness (comparison between treated groups and control group) of our driving simulation is valid in analyzing driving behaviors.”

Question 2. Lines 252 -255: Reading these lines and your response to my third question, it’s seems that you have used the “swarm” function of the SCANeR Studio, is it true? Unfortunately, this function is developed to guarantee a bubble of surrounding vehicles that constantly moves around the human driver, without considering any traffic law. Setting speed and time headway, does not guarantee specific traffic flows when “swarm” function is activated.

Response: Thank you for this comment. No, we didn’t use the ‘Swarm’ function of the SCANeR Studio. Vehicles controlled by the ‘Swarm’ function will disappear when they are far away from the test vehicle (controlled by participants) and appear on a random position around the test vehicle. So if we use this function, we can not control the surrounding environment as expected. So we use another way to design our scenario. There are about 27 vehicles in the scenario. In our scenario, there are only 3 lanes in one direction. The 27 vehicles are arrayed in 3 (lanes)*9(rows). The speed of the first vehicle on each lane is controlled to run in 40 or 70 km/h. For other vehicles, only their headway times are set, and they are not permitted to change lanes or overtake. The test vehicle is placed on the 7th row. When the scenario is running, these vehicles will result in a stable traffic flow state. When a car-following process is activated, only the vehicle in front of the test vehicle changes its speed according to the design.

Question 3. A driver involved in a crash at the ds should immediately stop driving and his participation in the experiment. This is because his driving behaviour could very likely be altered: fear, stress, excessive caution, unrealistic sensations, etc can normally occur.

Maybe is better: “If car-following progress is interrupted (by lane-change or overtaking), the corresponding data will be discarded in the extract of the car-following parameter.” instead of “If car-following progress is interrupted (by a crash or lane-change), the corresponding data will be discarded in the extract of the car-following parameter.”.

Response: Thank you for this comment. We have modified this statement.

Question 4. Response to my sixth question: Please, provide the number of effective car-following manoeuvres carried out by each driver.

Response: Thank you for this comment. The number of effective car-following maneuvers for each type of weather condition is 18 (=2 (traffic flow state except for free flow) * 3 (road types) * 3(car-following situation)).

Question 5. Line 295: “based on the gender composition of drivers” is not really clear. “based on the demographic characteristics of Chinese drivers” sounds better. Please provide a reference for this last statement.

Response: Thank you for this comment. We have modified this statement. The reference has been added (reference [30], http://www.niuche.com/news/detail_479869.html)

Question 6. Lines 459-462: The use of a driver database from which drawn your driver samples is only a partial solution to the issue of the driving simulator sickness because it depends by various factors, such as characteristics of driving scenario and psychophysics conditions of drivers. So, the probability that the same driver/subject became sick in a driving simulator varies over time and by type of experiments (length, complexity, driving environment, etc). Moreover, it’s mandatory to always monitor the level of simulator discomfort of each participants to avoid after-effects in the results. Therefore, I strongly recommend the authors to monitor simulator sickness in the future experiments.

Response: Thank you for this comment. The driving simulator sickness is surely a problem. In the past studies, we used a questionnaire that includes the comfort levels before/after driving to assess the psychophysics conditions. In the future experiments, we will pay more attention on monitoring simulator sickness. First, the questionnaire will be carefully designed. By collecting drivers’ psychophysics data during each rest, the driving data will be determined if it should be discarded in the following analysis. Besides, the driving time of one scenario will be well controlled to avoid the driving simulator sickness. The laboratory staff will be required to monitor driver status in real time.

Given the various limitations of experimental protocol, the partially convincing methodology and modest results, the article continues to be a little limited and not totally conclusive. However, a contribution, although minimal, to research literature is identifiable.
